

# The Urmia Playa as source of airborne dust and ice nucleating particles – Part 2: Unraveling the relationship between soil dust composition and ice-nucleation activity

Nikou Hamzehpour[1,2*], Claudia Marcolli[2], Kristian Klumpp[2], Debora Thöny[3], Thomas Peter[2]

[1]Department of Soil Science and Engineering, Faculty of Agriculture, University of Maragheh, Maragheh, Postal Box: 83111-55181, Iran.

[2] Department of Environmental Systems Science, Institute for Atmospheric and Climate Sciences, ETH Zurich, 8092 Zurich, Switzerland.

[3] Department of Chemistry and Applied Biosciences, ETH Zurich, Vladimir-Prelog-Weg1-5, CH-8093 Zurich, Switzerland.

*Correspondence to*: Nikou Hamzehpour (nhamzehpour@maragheh.ac.ir) or Claudia Marcolli (claudia.marcolli@env.ethz.ch)

## Abstract

Ice-nucleating particles (INPs) originating from deserts, semi-arid regions and dried lakebeds may cause heterogeneous ice nucleation impacting cloud properties. Recently, due to climate change and water scarcity, abandoned agricultural lands with little surficial crust and negligible vegetation cover have become an increasing source for atmospheric dust worldwide. Unlike deserts, these areas are rich in soluble salt and (bio-)organic compounds. Using soil samples from various sites of the Lake Urmia Playa (LUP) in northwestern Iran and airborne dusts collected at nearby meteorological stations we elucidate how minerals, soluble salts and organic matter interact to determine the IN activity of saline soils and dust. X-ray powder diffraction shows that the mineralogical composition is dominated by K-feldspars (microcline), quartz, carbonates, and clay minerals. The samples were stepwise stripped of organic matter, carbonates, and soluble salts. After each removal step, the ice nucleation (IN) activity was quantified in terms of onset freezing temperatures ($T_{het}$) and heterogeneously frozen fractions ($F_{het}$) by emulsion freezing experiments using differential scanning calorimetry (DSC). We examined the influence of soluble salts and pH on microcline and quartz in emulsion freezing experiments, and comparing these with reference suspensions of microcline and quartz, exposed to salt concentrations and pH characteristic of the LUP samples. These analyses, combined with correlations between $T_{het}$ and $F_{het}$, allow us to identify the components that contribute to or inhibit IN activity. The LUP dusts turn out to be very good INPs with freezing onset temperatures around 248 K in immersion freezing experiments. Interestingly, their IN activity proves to be dominated by the relatively small share of (bio-)organic matter (1–5.3 %). These organic INPs are rich in carbohydrates, cellulosic and proteinaceous compounds, as determined by thermogravimetric analysis coupled with mass spectrometry (TGA-MS). After organic matter removal, the remaining IN activity ($T_{het} \approx 244$ K) can be traced back to the clay fraction, because $T_{het}$ and $F_{het}$ correlate positively with the clay mineral content but negatively with quartz and microcline. We attribute the inability of quartz and microcline to act as INPs to the basic pH of the LUP samples as well as to the presence of soluble salts, which lower ice nucleation temperatures below the freezing point depression due to the salt-induced reduction in water activity. This is confirmed by the reference freezing experiments with quartz and microcline suspensions. After additionally removing soluble salts and carbonates, the IN activity of





the samples increases again significantly ($T_{het} \approx 249$ K) and the negative correlation with quartz and microcline turned into a slightly positive one, which we explain by the recovered active sites of these minerals. Removing carbonates and salts from the natural samples leads to an increase in $T_{het}$ and $F_{het}$ as well, indicating that their presence also suppresses the IN activity of the (bio-)organic INPs. Carbonates and soluble salts could do this, for instance, through occlusion of (bio-)organic INPs from the solution by a cementing effect, through their adsorption on the mineral surface, through interactions with dissolved $Ca^{2+}$ undergoing complexation with the organic matter, or a combination of these effects. Overall, this study demonstrates that mineral and organic INPs do not just add up to yield the IN activity of soil dust, but that the freezing behavior is governed by inhibiting and promoting interactions between the components.

**Key words:** Abandoned agricultural lands, ice nucleating particles, oxygenated organic carbon, soluble salts, microcline, quartz.

### 1 Introduction

The ice content in clouds determines key cloud properties such as albedo, lifetime, and precipitation formation (Lohmann et al., 2006; Field and Heymsfield, 2015; Mülmenstädt et al., 2015). An accurate modelling of ice formation in clouds is therefore key to estimate the Earth's radiation budget in response to climate change (IPCC, 2013; Lohmann and Neubauer, 2018). Ice formation through homogeneous nucleation occurs readily within the volume of cloud droplets at temperatures below about 237 K, yet, at higher temperatures, this process becomes too improbable to be of relevance, and seeds are required to initiate cloud glaciation. These seeds can be ice crystals falling from higher cloud layers to layers below through a seeder-feeder process (Purdy et al., 2005; Ramelli et al., 2021; Proske et al., 2021), or ice-nucleating particles (INPs), which contain sites that catalyze the formation of the ice phase (Kanji et al., 2017). The number of airborne particles that are able to induce ice at the highest temperatures is low but increases with decreasing temperature. Considerable effort has been undertaken to quantify the atmospheric INP population and its temporal fluctuation globally (e.g. Bigg, 1973; Kanji et al, 2017; Welti et al., 2020; Brunner et al., 2021). Measurements have been performed by means of online techniques on a single particle level, e.g. in continuous flow diffusion chambers (DeMott et al., 2010; Tobo et al., 2013; Lacher et al., 2018; 2021; Welti et al., 2018; 2020; Brunner et al., 2021), or through filter sampling with subsequent analysis of the IN activity directly on the filters (Schrod et al., 2017; Price et al., 2018) or in drop-freezing assays (O'Sullivan et al., 2018, Testa et al., 2021, Chen et al., 2021; Welti et al., 2018; 2020). In most of these studies, direct physicochemical characterization of samples is limited. Usually the type of INPs is indirectly inferred through air mass back trajectory analysis. In some of these studies, the sensitivity of IN activity to heat or $H_2O_2$ digestion is tested to infer the contribution of biogenic material to INPs (O'Sullivan et al., 2018; Testa et al., 2021). Some studies present an elemental characterization (Schrod et al., 2017; Welti et al., 2018; Price et al., 2018) or fluorescence measurements to infer the presence of biological INPs (Tobo et al., 2013).

Alternatively, samples have been surface-collected and subsequently analyzed in drop-freezing assays (Conen et al., 2011; O'Sullivan et al., 2014; Tobo et al., 2014; Hill et al., 2016; Suski et al., 2018), in cloud chambers (Steinke et al., 2016; 2020) or in continuous flow diffusion chambers (Boose et al., 2016; 2019; Paramonov et al., 2018). Here, samples are sufficiently abundant to perform more detailed analyses and physicochemical characterization. One major finding from these studies is that (bio-)organic material present in soil dusts enables them to freeze water at higher temperatures than pure mineral dust does. INPs that are active at temperatures above 255–258 K have been ascribed to biogenic material due to their heat lability (O'Sullivan et al., 2018), their



sensitivity to $H_2O_2$ digestion, and through correlation of IN activity with the organic share of the samples, or a combination of these methods (Conen et al., 2011; Tobo et al., 2014; Hill et al., 2016; Testa et al., 2021).

Below 255 K, the role of (bio-)organic material as source of INPs is unclear. Paramonov et al. (2018) found a decrease of IN activity in the temperature range 233–243 K after removing heat-sensitive material from soil samples yet, the decrease in IN activity was not proportional to the amount of organic matter present in the samples. The $H_2O_2$ treatment did not affect or even

enhanced IN activity of the investigated soil samples. Boose et al. (2019) found that the mineralogical composition determined the IN activity of most desert dust samples that they investigated at temperatures up to 242 K with the exception of one carbonaceous sample, where most of the IN activity stemmed from organic matter.

To gain direct insight into the IN activity of aerosol constituents, freezing experiments with surrogate samples for different types of aerosols have been performed. Minerals and biological materials have attracted most attention as they are considered the

dominant INP types below and above 255–258 K, respectively. In laboratory studies, diverse types of organic and biogenic material proved to be IN active. Biological INPs include whole bacteria, pollen, spores, but also fragments of plants and animals (Morris et al., 2013; Després et al., 2012; Kanji et al., 2017). Moreover, also organic macromolecules of biological origin proved to be IN active, such as proteinaceous material (Huang et al., 2021), humic and fulvic substances (Fornea et al., 2009: Wang and Knopf, 2011; Borduas-Dedekind et al., 2019), cellulose (Hiranuma et al., 2015a; 2019), polysaccharides (Steinke et al., 2020) and lignin

(Bogler and Borduas-Dedekind, 2020; Steinke et al., 2020).

Among the minerals present in atmospheric dust (Murray et al., 2012; Kanji et al., 2017), feldspars, clay minerals, and quartz are considered most relevant as INPs. Other common minerals like calcite, dolomite and micas proved to be hardly IN active. Among the feldspars, K-feldspars showed higher IN activity than (Na–Ca) feldspars (Zolles et al., 2015; Kaufmann et al., 2016; Harrison et al., 2016; 2019; Kumar et al., 2019b), and especially microcline remained IN active up to the highest temperatures observed for

mineral particles (Atkinson et al., 2013; Kaufmann et al., 2016; Welti et al., 2019). Yet, the IN activity of feldspars also proved to be highly sensitive to the presence of solutes: while low concentrations of ammonia and ammonium ions enhanced it, alkali ions and acids deteriorated it (Kumar et al., 2018; 2019b; Whale et al., 2018; Perkins et al., 2020; Yun et al., 2020). Reference quartz samples showed high IN activity, yet, only after being freshly milled. This indicates that IN active sites on quartz depend on its surface structure and are most probably related to defects (Zolles et al., 2015; Kumar et al., 2019b). Clay minerals dominate the

submicron fraction of atmospheric dust and can be transported over long distances and even between continents (Murray et al., 2012). All clay minerals that have been tested as INPs so far, that is kaolinite, montmorillonite and illite, proved IN active, yet, at lower temperatures than K-feldspars and quartz (Pinti et al., 2012; Hoose and Möhler, 2012; Hiranuma et al., 2015b; Kanji et al., 2017).

Only few studies have tried to relate the IN activity of natural samples with their mineralogical composition (O'Sullivan et al.,

2014; Kaufmann et al., 2016, Boose et al., 2016; Paramonov et al., 2018). Kaufmann et al. (2016) were able to explain the freezing behavior of most of their surface-collected dust samples by their mineralogical composition. Boose et al. (2016; 2019) found a correlation between the IN activity of dust samples sourced from different deserts and their quartz and feldspar content. Yet, some of the samples were milled prior to performing freezing experiments, which might have enhanced the IN activity of quartz (Zolles et al., 2015; Kumar et al., 2019a). Paramonov et al. (2018) found a good correlation of IN activity with total feldspar and K-feldspar

content of surface-collected dust. A relationship between mineralogical composition and IN activity was also observed by O'Sullivan et al. (2014) for temperatures below 255 K.



Different parameterizations to quantify the immersion freezing of dust particles have been proposed. Niemand et al. (2012) developed a parameterization of ice active site density per surface area from cloud chamber measurements of desert dust samples. The parameterization proposed by DeMott et al. (2015) is based on laboratory studies and atmospheric measurements of dust representative of Saharan and Asian desert sources and links the prediction of INP number concentration to particle number concentrations of sizes larger than 0.5 µm. These parameterizations premise that, to a first order, mineral dust may be assumed as one particle type. Alternatively, a specific mineral type is taken representative for the mineral dust fraction like K-feldspars (Atkinson et al., 2013) or illite (Hiranuma et al., 2015b). These parameterizations have the advantage of being simple, yet, they neglect the complexity of soil dust and potential aging effects in the atmosphere.

Soil types that expand due to human influence are dried lakebeds, abandoned agricultural land, and saline soils on the margin of drying lakes (Abuduwaili et al., 2010; Goodman et al., 2019; Sweeney et al., 2016; Varga et al., 2014; Perez and Gill, 2009; Washington et al., 2006; Prospero et al., 2002). Such soils contribute significantly to aerosols locally, regionally, and potentially globally, as exemplified by the enormous amounts of dust that are deflated annually from the Aral Sea in Kazakhstan and Uzbekistan and Ebi Nur Lake in northwestern China (Abuduwaili et al., 2010). In this study, we focus on desiccation of Lake Urmia (LU), one of the biggest salt lakes in the world in the northwest of Iran. Increasing soil salinity, which negatively affects plant growth and crop production, led to soil erosion and land degradation (Gorji et al., 2020; Jiang et al., 2019; Scudiero et al., 2015), and resulted in the remain of vast barren lands composed of fine textured, lacustrine sediments (Hamzehpour et al., 2018; Shadkam et al., 2016; Farokhnia and Morid, 2014). These saline areas with limited vegetation cover in most parts might be of high vulnerability to wind erosion due to their unique physicochemical characteristics. Recent studies have indeed demonstrated the increase in the intensity of dust storms over LUP and nearby cities (Ahmady-Birgani, 2020; Boroughani et al., 2019; Sotoudeheian et al., 2016).

In the companion paper of this work (Hamzehpour et al., 2022), we identified highly erodible playa surfaces along the western Lake Urmia Playa (LUP), and characterized four surface-collected soil samples together with airborne dust samples from nearby meteorological stations with respect to physicochemical properties, mineralogical composition, and IN activity. Emulsion freezing experiments in a differential scanning calorimeter (DSC) revealed that some of the lower concentrated suspensions of the dust samples (2 wt %) exhibited a higher heterogeneously frozen fraction ($F_{het}$) of droplets than the higher concentrated samples (5 wt %). Such a reciprocal proportionality points to interactions between sample constituents that inhibit IN activity. Moreover, we found a negative correlation of IN activity with K-feldspar and quartz content, which is opposite to findings from previous studies (O'Sullivan et al., 2014; Boose et al., 2016; 2019; Paramonov et al., 2018). To elucidate the potential contribution or inhibition of (bio-)organic matter, minerals, and soluble salts to the IN activity, we performed treatments to remove carbonates, soluble salts, and organic matter from the samples. To attribute the observed change in IN activity to specific interactions between soluble salts and minerals, we performed reference freezing experiments with quartz and microcline suspensions with salt concentration and pH in the typical range of the LUP samples. The insights gained from these treatments are the subject of this paper.

## 2 Material and Methods

### 2.1 Sampling locations

The sampling procedure has been described in detail in Part 1 of this work (Hamzehpour et al., 2022). Here, we recapitulate the relevant information for the present study. Four soil samples were collected from the top 5 cm surface layer of highly wind erodible areas from north to the south of Lake Urmia Playa (LUP). Together with the soil samples, airborne dust samples were collected at



meteorological stations nearby the soil sampling areas. Dust samples were collected using high-volume samplers manufactured by

Graseby–Andersen (Smyrna, Georgia, USA) on a 20.3 cm × 25.4 cm glass micro-fiber filter (Whatman Inc., USA) at flow rates of 1.13–1.41 m³. min⁻ for 24 h. Detailed discussion of the dust sources in the region are presented in the companion paper. Here, we just give a summary of the sampling locations in Table 1.

**Table 1.** The location of the collected dust samples and their corresponding soil samples

| Dust sample | Location (wrt to LU) | Longitude (E) | Latitude (N) | Corresponding soil sample (s) | Land type | Longitude (E) | Latitude (N) |
|---|---|---|---|---|---|---|---|
| Salmas (Sa) | north | 44º49'10" | 38º12'38" | Soil Sa | AAL | 45º01'17" | 38º13'27" |
| Jabal (Jab) | northwest | 45º01'34" | 37º51'44" | Soil Jab | Sa-Sh | 45º03'22" | 37º5036" |
| Merang (Mer) | west | 45º09'14" | 37º47'35" | Soil Mer | Sa-SC | 45º07'47" | 37º49'24" |
| Miandoab (MD) | south | 45º58'04" | 37º03'58" | Soil MD | SC-CF | 45º46'44" | 37º06'49" |

AAL: abandoned agricultural lands; Sa-Sh: sand sheets; Sa-SC: sandy salt crusts; SC-CF: salt crusts-clay flats.


**Table 2.** Physicochemical properties of the studied natural dust samples and surface collected soil samples.

| sample | EC (dS m⁻¹) | TC (%) | OM (%) | pH | Particle size classes (< 2mm) (%) | | |
|---|---|---|---|---|---|---|---|
| | | | | | Clay (<2 µm) | Silt (2–50 µm) | Sand (50–2000 µm) |
| Soil Sa | 0.9 | 21.1 | 5.3 | 8.2 | 15.8 | 20.2 | 64 |
| Dust Sa | 27.5 | 14.4 | 2.6 | 7.5 | 5.7 | 63.2 | 35.1 |
| Soil Jab | 35.3 | 28.5 | 1.0 | 8.6 | 1.8 | 6.9 | 91.2 |
| Dust Jab | 43.9 | 43 | 1.3 | 8.3 | 1.1 | 2 | 97 |
| Soil Mer | 40 | 19.1 | 3.3 | 8.2 | 9.3 | 59.2 | 31.6 |
| Dust Mer | 19.9 | 17.1 | 2.9 | 8.2 | 3.0 | 10.2 | 86.9 |
| Soil MD | 16.3 | 13.1 | 1.2 | 8.1 | 3.0 | 13.1 | 83.9 |
| Dust MD | 4.9 | 13.5 | 1.7 | 8.0 | 3.9 | 15.6 | 79.7 |

EC: electrical conductivity; TC: total carbonates; OM: organic matter.

### 2.2 Physicochemical analysis

A detailed characterization of the samples has been given and discussed in Part 1 of this work (Hamzehpour et al., 2022). Here,

we summarize the applied methods and replicate the findings with relevance for this study.

The soil samples were passed through a 2 mm sieve and, along with dust samples, their physicochemical properties were determined as summarized in Table 2. Soil electrical conductivity (EC) and acidity (pH) were measured in a 1:2.5 soil to water suspension using a Jenwey conductivity meter (model 4510) and VWR Symphony SB70P pH meter, respectively (Rhoades, 1996). In addition, soil organic carbon (OC) was measured using a wet oxidation technique (Nelson and Sommers, 1996) and converted





to organic matter (OM) by multiplying by a factor of two. Soil total carbonates were determined through back titration of the

remaining HCl (Page et al., 1982).

**Table 3.** Mineralogical composition of natural soil and dust samples in % derived from Rietveld analysis of the XRD patterns.

| minerals | Soil Sa | Dust Sa | Soil Jab | Dust Jab | Soil Mer | Dust Mer | Soil MD | Dust MD |
|---|---|---|---|---|---|---|---|---|
| Quartz | 14.3±0.3 | 20.9±0.4 | 23±0.3 | 32±0.4 | 14.9±0.2 | 14.6±0.3 | 31.1±0.4 | 27.5±0.4 |
| k-Feldspar (microcline) | 3.5±0.3 | 5.6±0.4 | 6.1±0.3 | 5.9±0.5 | 6.9±0.3 | 6.3±0.4 | 5.2±0.3 | 3.9±0.2 |
| Na-Plagioclase | 9.8±0.3 | 14.4±0.3 | 16.6±0.4 | 10.4±0.4 | 14.1±0.3 | 14.4±0.5 | 15.3±0.5 | 13.4±0.4 |
| Hornblende | 1.9±0.3 | 2.7±0.3 | 3.7±0.2 | 4.2±0.2 | 3.9±0.4 | 3.7±0.3 | 3±0.2 | 2.7±0.3 |
| Total silicates | 29.5 | 43.6 | 49.4 | 52.5 | 39.8 | 39 | 54.6 | 47.5 |
| Calcite | 13.5±0.3 | 12.5±0.3 | 10.8±0.2 | 11.2±0.2 | 9.7±0.3 | 9.8±0.3 | 10.1±0.2 | 10.3±0.2 |
| Dolomite | 3.2±0.2 | 1.9±0.2 | 4.3±0.2 | 4.1±0.2 | 4.5±0.3 | 3.9±0.3 | 3±0.3 | 3.2±0.2 |
| Aragonite | 4.4±0.3 | – | 13.4±0.3 | 8.7±0.3 | – | 4±0.2 | – | – |
| Magnesite | – | – | – | – | 4.9±0.3 | 3.4±0.2 | – | – |
| Total carbonates | 21.1 | 14.4 | 28.5 | 24 | 19.1 | 21.1 | 13.1 | 13.5 |
| Kaolinite | 8.4±0.6 | 6.1±0.7 | 2.2±0.5 | 2.2±0.3 | 4.9±0.6 | 3.3±0.6 | 1.6±0.4 | 7.2±0.2 |
| Smectite | – | – | – | – | – | – | 10.6±0.9 | 6.3±0.9 |
| Palygorskite | 12.3±1.0 | 9.8±0.8 | 3.8±0.6 | 3.8±0.6 | 12.6±0.7 | 11.7±0.9 | 4.1±0.5 | 5.5±0.6 |
| Chlorite | 16.8±0.9 | 9±0.9 | 9.6±0.6 | 8.2±0.8 | 11.8±0.9 | 10.3±0.9 | 3.9±0.4 | 4.4±1.2 |
| Total clay minerals | 37.5 | 24.9 | 15.6 | 14.2 | 29.3 | 25.3 | 20.2 | 23.4 |
| Mica | 11.1±0.4 | 9.2±0.4 | 4.9±0.2 | 6.9±0.5 | 8.7±0.3 | 8.3±0.4 | 7.4±0.3 | 9±0.3 |
| Serpentine | 0.8±0.2 | – | – | – | – | – | – | 0.2±0.2 |
| Total phyllosilicates | 49.4 | 34.1 | 20.5 | 21.1 | 38 | 33.6 | 27.6 | 32.6 |
| Gypsum | – | 3.2±0.2 | 0.3±0.1 | 0.7±0.3 | 0.6±0.1 | 0.6±0.1 | 2.8±0.2 | 4.4±0.2 |
| Halite | – | 4.7±0.1 | 1.3±0.1 | 1.7±0.1 | 2.5±0.1 | 5.7±0.1 | 0.7±0.1 | 2±0.1 |
| Magnetite | – | – | – | – | – | – | 1.2±0.1 | – |

### 2.3 Quantitative mineralogy

The mineralogical composition of the bulk dust samples and surface collected soil samples was investigated with X-ray diffraction

(XRD) analysis (Bish and Plötze 2010) followed by quantitative Rietveld analysis. More details are given in Part 1 (Hamzehpour

et al., 2022). We replicate the results in Table 3.

### 2.4 Thermogravimetric analysis

Thermogravimetric analysis (TGA) and differential scanning calorimetry (DSC) of soil and dust samples smaller than 63 µm were

measured on a STA 449 F5 Jupiter from NETZSCH coupled to a QMS 403 D Aëolos mass spectrometer. The gas flow was set to

60 + 20 mL Ar/min and 20 mL $O_2$/min to obtain an atmospheric mixture. A soil or dust sample (10–20 mg) was placed in an $Al_2O_3$



crucible and heated up with a heating rate of 10ºC min⁻¹ from 40ºC to 1000ºC. In TGA thermograms, the region 120–190 ºC is a transition between endothermic water loss and exothermic organic matter oxidation. The data was analyzed and plotted with the Proteus software from Netzsch.

**2.5 Ice nucleation activity of treated soil and dust samples**

A statistical design with six treatments, each with one replication was used to study the effects of soluble salt removal (SR), carbonates removal (CR) and organic matter removal (OMR) on IN activity of soil and dust samples as illustrated in Fig. 1. Untreated samples (natural samples) are used as the reference. Apart from the single treatments, combined treatments of salt and carbonate removal (SR + CR), salt and organic matter removal (SR + OMR), and carbonate, salt, and organic matter removal (CR

+ SR + OMR) were performed. The combined organic matter and carbonates removal (OMR+CR) was excluded from further analysis because the measurement of electrical conductivity of the thus treated samples revealed a concomitant major loss of salts, making them similar to the combined CR + SR + OMR treated samples.

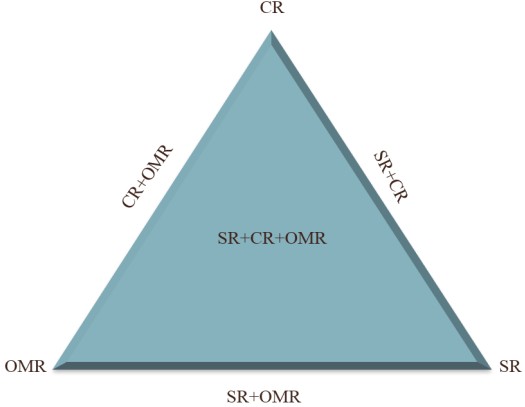

**Figure 1.** Diagram of treatments used to investigate the role of soil soluble salts, carbonates and organic matter on IN activity of natural soil and
dust samples from LUP. SR: salt removed; CR: carbonate removed; OMR: organic matter removed.

To remove stepwise soluble salts, carbonates and organic matter from the soil and dust samples, we followed the procedure described by Jackson (1985), Kittrick and Hope (1963), and Dane and Topp (2020) as outlined below.

**2.5.1 Removal of soluble salts**

In order to remove soluble salts, samples were washed with distilled water. The samples were suspended in distilled water and
shaken for 1 min, followed by centrifugation with 6000–7000 rpm for 5 min. A ratio of 1:2.5 soil to water was found to be efficient enough for removing salts after examining different ratios in test experiments. The procedure was repeated until the leached salt concentration dropped below 10 mM. After drying in an oven at 60ºC, the samples were stored in 20 mL glass vials for further analysis.

**2.5.2 Removal of carbonates**

Carbonates were removed from the samples by using a 1 M sodium acetate buffer solution with adjusted pH of 5. Careful pH monitoring is important because for pH > 5, dissolution of carbonates remains incomplete, while pH < 5 can lead to the dissolution





of iron hydroxides. 40 mL of sodium acetate buffer solution was added to 1000 mg of each sample in a 50 ml Erlenmeyer flask and then kept at 60°C in a water bath until $CO_2$ bubbles were no longer visible. Then, the suspension was centrifuged and the supernatant decanted. In order to prevent organic matter decomposition of the samples, the temperature should not exceed 60°C.

This procedure was repeated until all carbonates were removed. To test whether this was the case, a small fraction of the sample was treated with 10 % HCl. If no bubbles developed, the samples were considered to be carbonate free. To remove all sodium acetate, the centrifuged and decanted samples were washed with distilled water, followed by centrifugation and decantation. The electrical conductivity of the samples was checked to make sure that the EC of the samples did not fall below the value of the untreated samples. Note that through this procedure the salt composition in the sample is altered while the electrical conductivity

is preserved. Finally, samples were dried in an oven at 60°C and stored in 20 mL glass vials for further analysis.

### 2.5.3 Removal of organic matter

In order to remove the organic matter from the samples, 5 mL of 30 % (wt/wt) $H_2O_2$ was added to a known amount of the samples in a tall 500 ml beaker and stirred until the reaction slowed. Then the suspension was placed in a water bath at 65–70°C until the reaction ceased and the solution became clear. $H_2O_2$ solution was added in 5 mL portions until no more reaction was evident.

Thereafter, the samples were centrifuged for 15 min in order to remove the supernatant. In case of sample dispersion, few droplets of 0.5 M $MgCl_2$ were added to the sample. Finally, the samples were dried in the oven at 60°C and stored in 20 mL glass vials for further analysis.

### 2.6 Reference experiments with the minerals microcline and quartz

In order to investigate the role of salt type and concentration, and pH on the IN activity of microcline and quartz, we performed

experiments with a microcline sample from North Macedonia (the same as has been used in Klumpp et al., 2022), and a quartz sample from Sigma Aldrich (the same as has been used in Kumar et al., 2019a) with 2 wt % and 1 wt % concentrations, respectively. NaCl, $CaCl_2$, and $MgCl_2$ (all Sigma Aldrich) were chosen as soluble salts, since they are the dominant salts in the sediments of LUP. Concentrations of 0.01 M, 0.1 M, and 0.4 M were chosen to cover the salinity range of the LUP samples. To investigate the combined effect of soluble salts and pH on the IN activity of quartz and microcline, each salt solution was adjusted to pH 5 (acidic

soils without carbonates); pH 7 (neutral) and pH 8 (representative of carbonate soils) by adding small portions of dilute NaOH or HCl solutions. Figure 2 illustrates the combinations of salt type, salt concentration, and pH used to perform the immersion freezing experiments with the reference microcline and quartz samples.

### 2.7 Freezing experiments

To characterize the average ice nucleation efficiency of both, natural and treated soil and dust samples, emulsion freezing

experiments with a differential scanning calorimeter (DSC) Q10 from TA instruments were performed. The <63 µm fraction was suspended in water (molecular bioeagent water, Sigma Aldrich) at concentrations of 2 wt % and 5 wt % and sonicated for 5 min in order to minimize particle aggregation. Emulsions were prepared by combining the suspensions with a mixture of mineral oil and lanolin (both Sigma Aldrich) at a ratio of 1:4 and stirring the mixture with a rotor stator homogenizer (Polytron PT 1300 D with a PT-DA 1307/2EC dispersing aggregate) for 40 s at 7000 rpm. For DSC measurements, 5 to 10 mg of emulsion were placed

in an aluminum pan, and the pan was hermetically sealed. Thermograms were registered at a rate of 1 K min$^{-1}$ in the temperature ranges of freezing and melting and evaluated in terms of the onset temperatures of the heterogeneous ($T_{het}$) and homogeneous freezing peaks ($T_{hom}$), the heterogeneously frozen fraction ($F_{het}$) and the melting temperature $T_{melt}$ as has been explained in more



detail in Kumar et al. (2018) and Klumpp et al. (2022). The stability of the emulsions was tested for some samples by subjecting them to three freezing cycles with a first and third cycle performed with a cooling rate of 10 K min$^{-1}$ as control cycles following

the procedure introduced by Marcolli et al. (2007). Before each experiment, emulsions were freshly prepared. Experiments were repeated at least once with a freshly prepared suspension.

The same procedure was used to investigate the IN activity of the reference minerals (microcline with 2 wt % and quartz with 1 wt % concentrations) in combination with different solute compositions as described in Sect. 2.6.

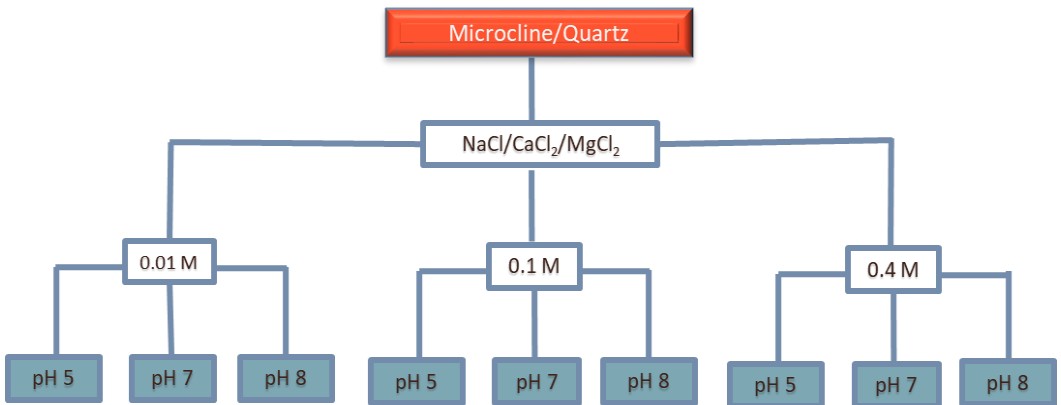

**Figure 2.** Statistical design of solution composition to investigate the role of salt type, salt concentration, and pH on the IN activity of microcline and quartz samples. The experiments were performed for suspension concentrations of 2 wt % for microcline and 1 wt % for quartz in solutions of the three salts NaCl, CaCl$_2$, and MgCl$_2$, with each salt at the three indicated concentrations, and with each concentration at three different pH values.

**3 Results and Discussion**

**3.1 Treated natural soil and dust samples**

To investigate the impact of soluble salts, carbonates and organic matter on the IN activity in immersion mode of the soil and dust samples Sa, Jab, Mer, and MD, we removed salts (SR treatment), carbonates (CR treatments) and organic matter (OMR treatments) from the samples in a systematic way as illustrated in Fig. 1. DSC thermograms of the treated samples at suspension concentrations of 2 wt % and 5 wt % were recorded and are displayed together with the thermograms of the untreated samples in Fig. 3 (dust

samples) and Fig. A1 in the Appendix (soil samples). $T_{het}$ and $F_{het}$ retrieved from the natural and treated samples are summarized in Table 4. In Figs. 4 and 5, effects of different treatments as $\Delta T_{het\ (treated-natural)}$ and $rF_{het\ (treated/natural)}$ of each investigated dust and soil sample at 5 wt % concentrations are presented. Triangle plots of $\Delta T_{het\ (treated-natural)}$ and $rF_{het\ (treated/natural)}$ for 2 wt % concentrations are shown in Figs. A2 and A3 in the Appendix. Finally, Fig. 6 summarizes the effects of all treatments.



**Figure 3.** DSC thermograms of treated dust samples with 5 wt % (left panels) and 2 wt % (right panels) concentrations in comparison with the untreated samples. SR: salt removed; CR: carbonate removed; OMR: organic matter removed; SR + CR: salt and carbonate removed; SR + OMR: salt and organic matter removed; SR + CR + OMR: salt, carbonate and organics removed. $T_{het}$ in Kelvin and $F_{het}$ in fraction are displayed directly on the curves.





**Table 4.** Mean $T_{het}$ and $F_{het}$ values for natural soil and dust samples and each treatment at 2 wt % and 5 wt % concentrations.

|  | Conc. (wt %) | natural | SR | CR | OMR | SR + CR | SR + OMR | SR + CR + OMR |
|---|---|---|---|---|---|---|---|---|
| $T_{het}$ (K) | 2 | 247.4±1.4 | 248.6±0.9 | 249.0±1.7 | 246.7±1.1 | 249.1±1.4 | 247.4±1.3 | 247.7±0.83 |
|  | 5 | 248.2±1.3 | 248.9±1.1 | 249.0±1.6 | 246.6±1.1 | 249.3±2.2 | 246.9±1.3 | 248.6±0.21 |
|  | mean | 247.8 | 248.8 | 249.0 | 246.7 | 249.2 | 247.1 | 248.1 |
| $F_{het}$ (%) | 2 | 0.62±0.11 | 0.74±0.05 | 0.77±0.07 | 0.64±0.06 | 0.79±0.04 | 0.70±0.09 | 0.70±0.04 |
|  | 5 | 0.63±0.12 | 0.77±0.06 | 0.78±0.06 | 0.64±0.1 | 0.80±0.05 | 0.64±0.1 | 0.74±0.06 |
|  | mean | 0.625 | 0.755 | 0.775 | 0.640 | 0.795 | 0.670 | 0.720 |


### 3.1.1 Salt removed (SR) samples

DSC thermograms of the samples after SR treatment are shown in Fig. 3 and Fig. A1 in the Appendix, in dark orange color. For almost all samples, removing soluble salts increased the IN activity for both suspension concentrations (see Figs. 4, 6a and Fig. A2 in the Appendix): $T_{het}$ increased by $\Delta T_{het\ (SR-natural)} = 1.0$ K from a mean value of 247.8 K to 248.8 K (Table 4). In all samples

with the exception of Soil Sa at both 5 and 2 wt % concentrations (Fig. 5 and Fig. A3 in the Appendix), $F_{het}$ increased after SR treatment (see Fig. 6b). The highest $rF_{het\ (SR/natural)}$ value (1.69) was observed for Dust Jab at 5 wt % concentration ($F_{het} = 0.78$ after SR, versus 0.46 in the natural sample), which is the sample with the highest electrical conductivity (EC) of 43.8 dS m$^{-1}$ (Table 2). The smallest difference in IN activity before and after SR treatment was observed for Soil Sa, the sample with the lowest soil EC (0.85 dS m$^{-1}$). Therefore, removing salts from this sample had almost no or even a negative effect on both $T_{het}$ ($\Delta T_{het\ (SR-natural)} = -$

0.16 K as the average of 5 wt % and 2 wt % samples) and $F_{het.}$ ($rF_{het\ (SR/natural)} = 1.0$).

In Part 1 of the manuscript (Hamzehpour et al., 2022) we showed that there was a significantly negative correlation between EC as an indicator of the presence of soluble salts and $T_{het}$ and $F_{het}$ of the 5 wt % samples (-0.68 and -0.75, respectively), implying that high soluble salt content had a negative influence on IN activity. The increase in $T_{het}$ and $F_{het}$ after SR treatments confirms the inhibiting effect of soluble salts on the IN activity of the investigated soil and dust samples at these concentrations.

### 3.1.2 Carbonate removed (CR) samples

DSC thermograms of the samples after carbonate removal are shown in Fig. 3 and Fig. A1 in the Appendix in black color. Similar to SR treatment, in almost all of the samples the CR treatment increased the IN activity for both concentrations (Figs. 4, 6a and Fig. A2 in the Appendix). $T_{het}$ increased on average by 1.2 K after carbonate removal (Table 4). The highest $\Delta T_{het\ (CR-natural)}$ was measured for Soil Sa (4.1 K for 2 wt % concentration), while the lowest one occurred in Soil MD with a slightly negative value of

-0.2 K at 2 wt % concentration (Fig. A1 in the Appendix). This sample also exhibits one of the lowest total carbonate content (TC from back titration (Table 2): 13.1 %; from mineralogical composition (Table 3): 13.1 %; from TGA-MS (Table 5): 11.95 %).
The CR treatment led to $rF_{het\ (CR/natural)} > 1$ for all samples (Fig. 5, 6b and Fig. A3 in the Appendix). The highest $rF_{het\ (CR/natural)}$ was observed in Soil Jab at 2 wt % concentration ($rF_{het\ (CR/natural)} = 1.78$), which exhibits one of the highest TC contents (28.5 %) and the lowest value belonged to Dust MD at 2 wt % ($rF_{het\ (CR/natural)} = 0.99$), which has one of the lowest TC contents (13.5 % in Table

285 2).
In Part 1 (Hamzehpour et al., 2022) of the manuscript we showed that $F_{het}$ of the natural samples are negatively correlated to TC and pH. The increase in $F_{het}$ after carbonate removal confirms this negative correlation.





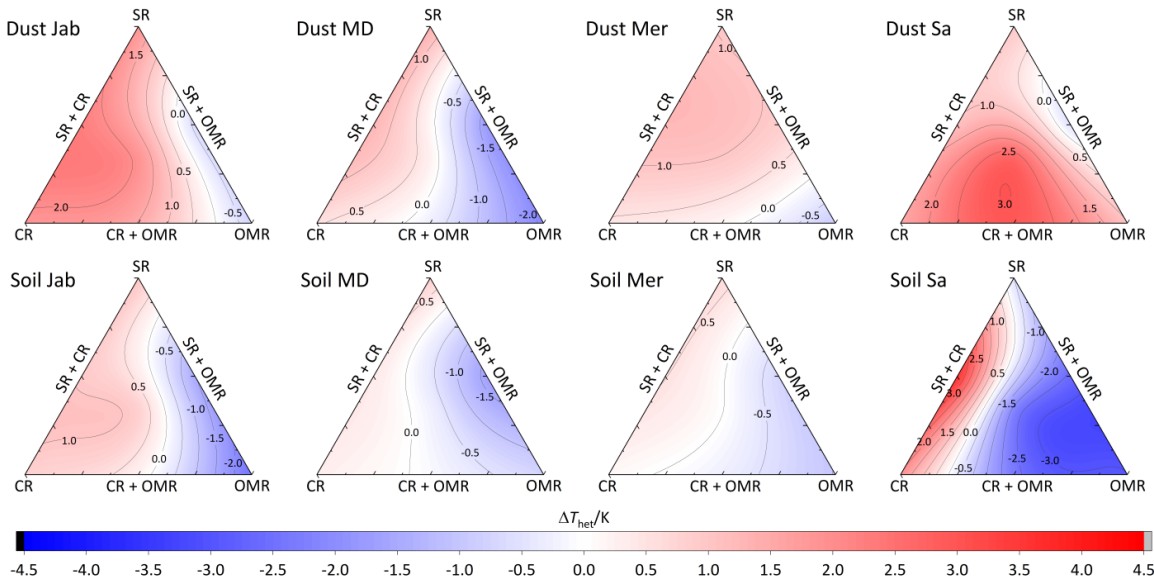

**Figure 4.** Effects of treatments in terms of $\Delta T_{het\,(treatment-natural)}$ for dust and soil samples at 5 wt % concentration. SR: salt removed; CR: carbonate removed; OMR: organic matter removed; SR + CR: salt and carbonate removed; SR + OMR: salt and organic matter removed; SR + CR + OMR: salt, carbonate and organic removed treatments.

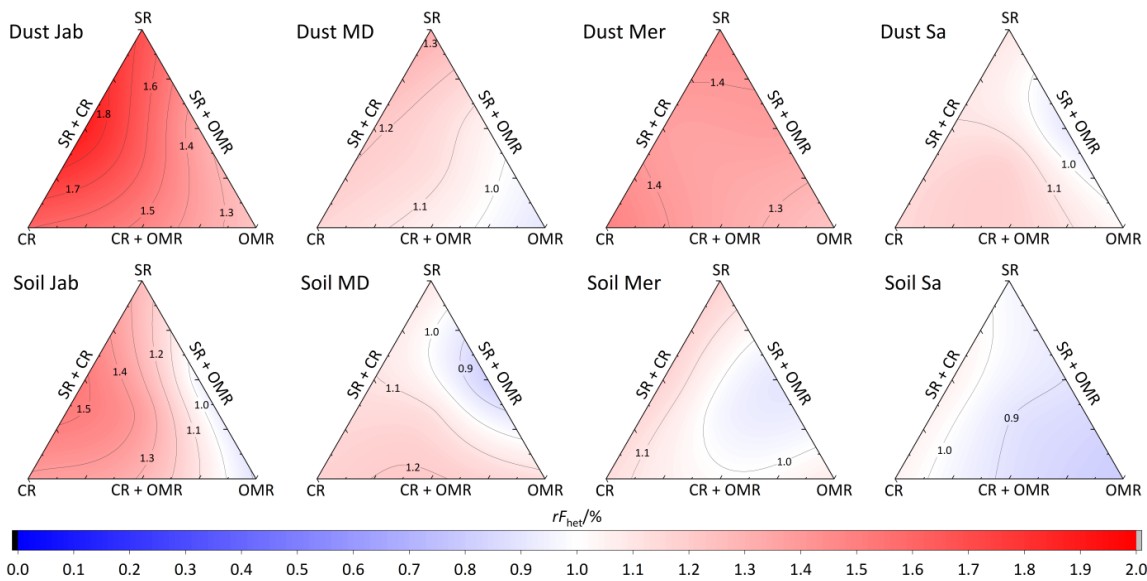

**Figure 5.** Effects of treatments in terms of $rF_{het\,(treatment/natural)}$ for soil and dust samples at 5 wt % concentration. SR: salt removed; CR: carbonate removed; OMR: organic matter removed; SR + CR: salt and carbonate removed; SR + OMR: salt and organic matter removed; SR + CR + OMR: salt, carbonate and organic removed treatments.



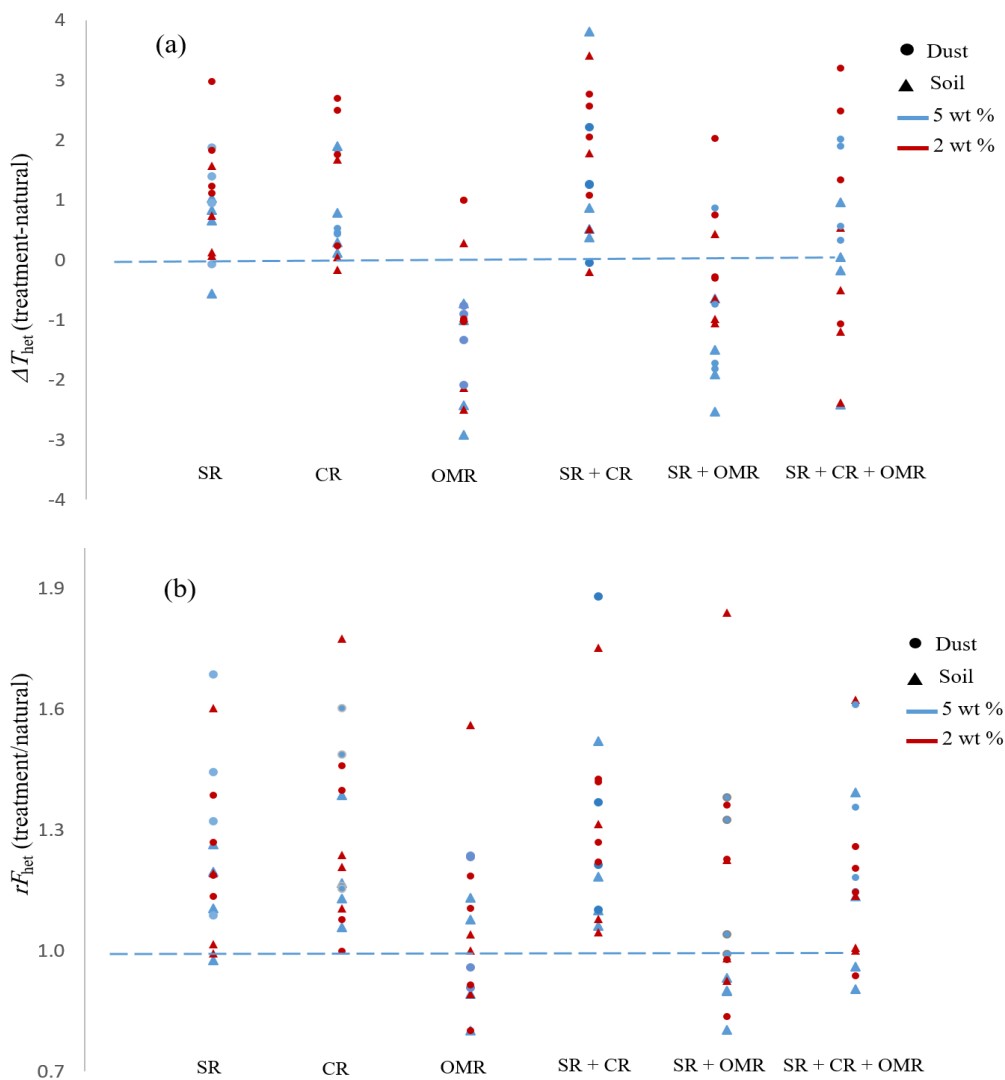

**Figure 6.** (a) Effect of treatments on $T_{het}$. Points above 0 (shown as dashed lines) represent the increase in $T_{het}$ after the treatment. (b) Relative change in $F_{het}$ after the treatments. Points above 1 show an increase in $F_{het}$ of the treated samples compared to the natural samples.

### 3.1.3 Salt and carbonate removed samples (SR + CR)

DSC thermograms of the samples after combined salt and carbonate removal are shown in Fig. 3 and Fig. A1 in the Appendix in blue color. $T_{het(SR+CR)}$ was on average 249.2 K, i.e. 1.4 K higher than for the natural samples (Table 4). Thus, the increase in $T_{het(SR+CR)}$ after the combined treatment is higher than the ones of each single treatment (Fig. 6a). Soil Sa exhibited the highest increase in $T_{het}$ after combined salt and carbonate removal with $\Delta T_{het\ ((SR+CR)-natural)} = 3.8$ K for the 5 wt % sample and 3.5 K for 2 wt % (Fig. A1 in thr Appendix), although its EC is negligible (0.9 dS m$^{-1}$) and TC (21.1 %) is average (Table 2). Comparison with


$T_{het}$ after carbonate removal reveals that this treatment alone also led to a strong increase in $T_{het}$ by $\Delta T_{het\ (CR\text{-}natural)} = 1.9$ K for the 5 wt % and even 4.1 K for the 2 wt % sample. Thus, we identify a strong interference of carbonates with the IN active species in this sample.

$F_{het}$ increased due to salt and carbonate removal for all samples (Fig. 6b). The highest $rF_{het\ ((SR+CR)/natural)}$ was observed in Dust Jab (Fig. 4) with a value of 1.9 K for the 5 wt % sample. Simultaneously, $T_{het}$ increased by $T_{het\ ((SR+CR)\text{-}natural)} = 2.2$ K for the 5 wt % sample and even by $T_{het\ ((SR+CR)\text{-}natural)} = 2.8$ K for the 2 wt % sample. Indeed, Dust Jab exhibits the highest EC (43.9 dS m$^{-1}$) and TC values (43 %) of all samples, and the second highest pH value (8.3). This again confirms that the presence of both carbonates and soluble salts reduce IN activity.

**3.1.4. Organic matter removed samples (OMR)**

DSC thermograms taken after organic matter removal, given in red color in Fig. 3 and Fig. A1 in the Appendix, show that this treatment reduced the freezing onset temperatures of most samples (blue colored region of the triangles in Figs. 4 and 5 and Figs. A2 and A3 in the Appendix). For 5 wt % concentration, $\Delta T_{het\ (OMR\text{-}natural)}$ was negative for all samples with an average decrease of 1.5 K. For the 2 wt % samples, $\Delta T_{het\ (OMR\text{-}natural)}$ ranged between -2.5 K and 1 K, with an average of $\Delta T_{het\ (OMR\text{-}natural)} = -0.7$ K. Soil

Sa, which exhibited the strongest shift to lower temperature after organics removal ($\Delta T_{het\ (OMR\text{-}natural)} = -2.9$ K), has the highest organic matter content among all studied samples (5.28 %). This demonstrates that freezing occurring at the highest temperatures strongly depends on the presence of organic matter and confirms the positive correlation (correlation coefficients $cc = 0.79$ for 5 wt % and $cc = 0.44$ for 2 wt % samples) between organic matter and freezing onset temperatures that had been reported in Part 1 (Hamzehpour et al., 2022). The dependence of $T_{het}$ on the presence of organic matter is also in agreement with various studies,

which showed that the IN activity of collected soil dust samples decreases when the samples are treated with heat or hydrogen peroxide to remove organics from soil dust samples (Conen et al., 2011; Tobo et al., 2014; Hill et al., 2016).

Although the correlation coefficient between $F_{het}$ and OM was significantly positive (0.77 for 5 wt % and 0.74 for 2 wt % concentration, Hamzehpour et al., 2022), $F_{het}$ did not decrease after organics removal with $rF_{het\ (OMR/natural)} = 1.06$ for 5 wt % and 1.04 for 2 wt % samples. For both concentrations, almost half of the samples showed an increase in $rF_{het\ (OMR/natural)}$ (Figs. 5 and

6b). In Soil and Dust Mer, $F_{het}$ increased due to OMR treatments (Fig. A1 in the Appendix) despite their rather high OM content (3.32 and 2.89 %, respectively). Therefore, it seems that the total content of organic matter is not the determining factor for the heterogeneously frozen fraction of the natural samples. Moreover, organic molecules may also have a hampering effect on the IN activity of the minerals, e.g. through adsorption on the mineral surfaces and blocking of IN active sites (Kleber et al., 2021).

The combined removal of soluble salts and organic matter shows similar trends in IN activity as the removal of OMR alone, but

less pronounced. For the 5 wt % samples, $T_{het}$ decreased on average by -1.3 K after combined SR + OMR treatment compared with -1.5 K for the OMR treatment alone. For the 2 wt % samples, the decrease in $T_{het}$ was -0.7 K for organic matter removal alone and reduces to only -0.04 K for combined salt and organic matter removal. Moreover, the few samples that showed an increase in $T_{het}$ after OMR treatment alone also did so for the combined SR + OMR treatment with Dust Mer at 5 wt % concentration as the only exception. Likewise, the combined SR + OMR treatment had a similar effect on $F_{het}$ as the organic matter removal alone, with a

few exceptions as can be seen in Figs. 6 and 8. Thus, the combination of salt and organic matter removal seems to be dominated by the effect of organics removal.

The correlation of $T_{het}$ with organic matter content and the decrease of $T_{het}$ after H$_2$O$_2$ digestion are strong evidence that the minor fraction of organics present in the soil and dust samples dominates freezing onset temperatures. To gain further insight into the chemical composition of the organic matter present in the samples, we performed TGA-MS coupled with DSC. The thermograms





show a stepwise decrease of mass (Fig. 7) that we can attribute to $H_2O$ or $CO_2$ loss as registered by the concurrently measured mass spectra. Furthermore, we can attribute mass loss to exothermic and endothermic processes based on the sign of the heat flow registered by DSC (Table 5, Figure A4 in the Appendix).

The weight loss up to about 250°C is attributable to loss of sorbed water. Dehydration of gypsum occurs from 100 to 150°C and is accompanied by an endothermic heat flow. Water bonded to smectites is also expected to be released in this temperature range

(Jananee et al., 2021). Combustion of soil organic matter can be identified through $CO_2$ release together with an exothermic heat flow. It exhibits two main steps of weight loss in the range from 250–550°C. The first, between 250 and 350°C, is associated with more easily oxidizable compounds, including simple protein molecules, polysaccharides (e.g., cellulosic material) and aliphatic structures. The second, between 350 and 550°C, is due to thermal degradation of recalcitrant, aromatic structures including lignin and non-hydrolysable compounds (Giannetta et al., 2018). Weight loss above 600°C is associated with the decomposition of

carbonates and characterized by an endothermic heat flow.

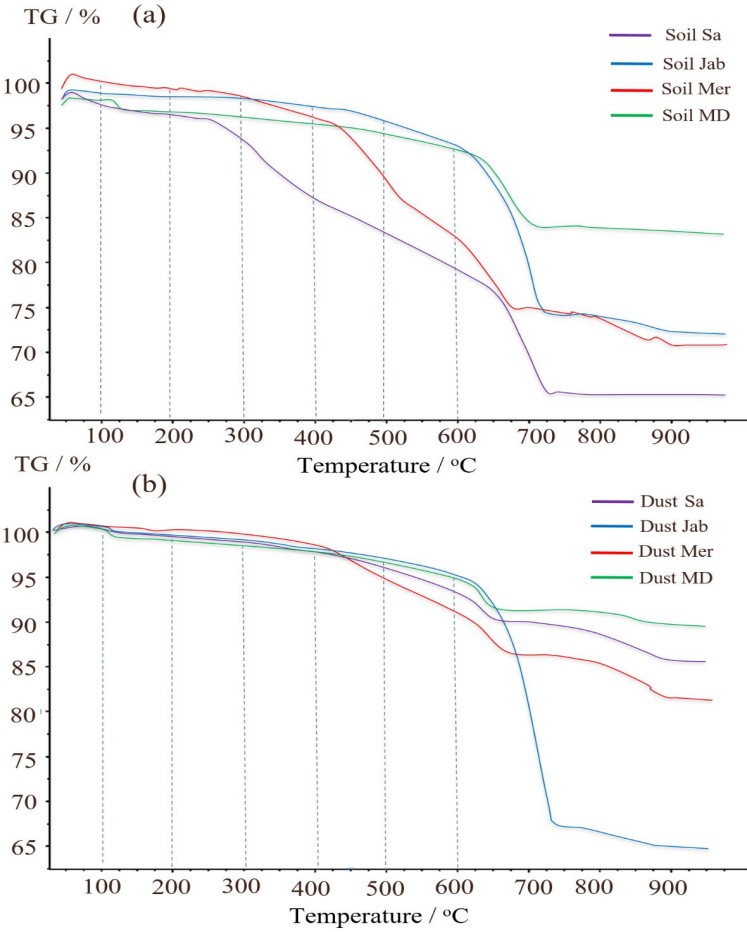

**Figure 7.** (a) TGA thermographs for soil and (b) for dust samples. Dashed lines delimit weight loss resulting from $H_2O$ (up to 200°C), organic compounds (200–550°C) separated into easily oxidizable (200–350°C) and recalcitrant (350–550°C) substances, and carbonate compounds (> 600°C).



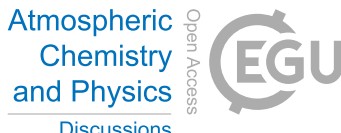

**Table 5.** Weight loss in % quantified with TGA-MS coupled with DSC thermograms in terms of different volatile and oxidizable fractions.

| Sample | 40–200°C H$_2$O | 200–350°C EO-OC | 350–550°C TS-OC | 250–550°C TOC | 550–750°C CaCO$_3$ | > 750°C IM |
|---|---|---|---|---|---|---|
| Soil Sa | 1.56 | 1.42 | 1.16 | 2.58 | 18.95 | 0.27 |
| Soil Jab | 0.43 | 0.17 | 0.51 | 0.68 | 26.07 | 1.36 |
| Soil Mer | 0.9 | 0.44 | 1.91 | 2.35 | 12.52 | 2.34 |
| Soil MD | 0.89 | 0.14 | 0.44 | 0.58 | 11.95 | 0.53 |
| Dust Sa | 1.12 | 0.61 | 0.89 | 1.5 | 10.64 | 4.96 |
| Dust Jab | 0.72 | 0.39 | 0.92 | 1.31 | 64.91 | 3.00 |
| Dust Mer | 0.78 | 0.41 | 1.47 | 1.88 | 16.34 | 5.55 |
| Dust MD | 1.19 | 0.27 | 0.73 | 1 | 10.45 | 1.94 |

OC: organic carbon; EO-OC: easily oxidizable OC; TS-OC: thermally stable OC; TOC: total organic carbon; IM: inorganic materials other than carbonates.

Overall, OC determined through TGA-MS is similar and correlates well ($cc = 0.93$) with OM determined through wet oxidation displayed in Table 2. According to Table 5, the ratio of EO-OS and TS-OC varies between samples. Soil Sa and Soil Mer had the highest total organic carbon (TOC) content among samples, however, the easily oxidizable fraction was highest in Soil Sa (1.42 %) while the thermally more stable fraction was largest in Soil Mer (1.91 %). Overall, IN activity of the soil and dust samples correlates better with the easily oxidizable organic fraction ($cc = 0.47$ with $T_{het}$ and 0.61 with $F_{het}$ as the average of 2 wt % and 5 wt % samples) than with the thermally more stable organics ($cc = 0.22$ with $T_{het}$ and 0.28 with $F_{het}$). Therefore, carbohydrates such as cellulose and proteinaceous molecules are the more likely contributors to IN activity of the investigated dust and soil samples than thermally more stable organics like lignin. Interestingly, the positive correlation between EO-OC increases after carbonate removal to $cc = 0.83$ with $T_{het}$ and 0.65 with $F_{het}$, pointing to an inhibiting effect of carbonates on the IN activity of these organics. Such an inhibiting effect seems to be especially strong for Soil Sa with $\Delta T_{het\ ((CR)-natural)} = 1.9$ K and $\Delta T_{het\ ((SR+CR)-natural)} = 3.8$ K for the 5 wt % and $\Delta T_{het\ ((CR)-natural)} = 4.1$ K and $\Delta T_{het\ ((SR+CR)-natural)} = 3.5$ K for 2 wt % concentration. This sample exhibits a strong decrease of $T_{het}$ after OMR treatment, which indicates that its onset freezing temperature is determined by the organic fraction. Thus, as the IN activity of the organic material is reduced in the presence of carbonates, there seem to be interactions between these species.

### 3.1.5 Salt, carbonate and organic matter removed samples (SR + CR + OMR)

DSC thermograms of the samples after SR + CR + OMR treatments are shown in Fig. 3 and Fig. A1 in the Appendix in purple color. In Figs. 4 and 5 and Figs. A2 and A3 in the Appendix, the centers of the triangles represent this treatment. With a value of 248.1 K, the mean $T_{het(SR+CR+OMR)}$ is only by 0.35 K higher (Table 4) than the mean $T_{het\ (natural)}$ (247.8 K), but lower than $T_{het}$ after SR, CR and SR + CR treatments. Yet, for the individual samples the trends are diverse: for three samples (Soil Sa, Soil Mer, and Soil MD), $\Delta T_{het\ ((SR+CR+OMR)-natural)}$ is negative at both concentrations (2 wt % and 5 wt %); for Dust MD, it is negative for 2 wt % and slightly positive for 5 wt % suspensions; for the other samples, it is positive at both concentrations. The highest decrease compared to the natural sample was in Soil Sa (5 wt %) with a 2.4 K reduction (Fig. 4 and Fig. A1 in the Appendix). The highest increase in $T_{het}$ was for the 2 wt % sample of Dust Jab with $\Delta T_{het\ ((SR+CR+OMR)-natural)} = 3.2$ K.



In most samples, $F_{het}$ increased after combined SR + CR + OMR treatments. Yet, for some samples with a negative $\Delta T_{het}$ $_{((SR+CR+OMR)-natural)}$, the decrease in $T_{het}$ was accompanied by a slight decrease in $F_{het}$ (Figs. 3, 6 and Fig. A1 in the Appendix). The strongest increase was for the 2 wt % sample of Soil Jab with $rF_{het\ ((SR+CR+OMR)/natural)}$ = 1.62 and for the 5 wt % sample of Dust Jab with $rF_{het\ ((SR+CR+OMR)/natural)}$ = 1.61. This strong increase is in accordance with their high EC and TC values together with their rather low organic matter content. These samples exhibit both rather high microcline and quartz contents such that the positive effects of removing salt and carbonates could have compensated the negative effect of organics removal, resulting in a net increase in heterogeneously frozen fraction.

Overall, the negative correlation between $T_{het}$ and microcline in natural samples ($cc$ of -0.46 and -0.28 for 2 wt % and 5 wt % samples, respectively) reverted to a slightly positive one after SR + CR + OMR treatment ($cc$ of 0.30 and 0.32 for 5 wt % and 2 wt % samples, respectively). For quartz, the negative correlation with $T_{het}$ also reverted to a positive one for the 5 wt % samples ($cc$ = 0.24), but remained negative for the 2 wt % samples. Moreover, the significantly positive correlation between $T_{het}$ and the total clay mineral content ($cc$ = 0.79) reversed to negative ($cc$ = -0.22) for the 5 wt % samples and remained only slightly positive for the 2 wt % samples ($cc$ = 0.17) but with a much lower value than for the natural samples ($cc$ = 0.65). Interestingly, for OMR treatment alone the correlation between $T_{het}$ and clay minerals was almost as high as for the natural sample ($cc$ of 0.62 and 0.71 for 5 wt % and 2 wt %, respectively), indicating that only the combined SR + CR + OMR treatment destroys the positive correlation. This again shows how intricate the interactions between the different soil and dust components are, and that the observed IN activity is the product of all these interactions.

### 3.2 Effect of salt, salt concentration and pH on IN activity of microcline and quartz

To substantiate further the impeding effect of soluble salts and carbonates on the IN activity of microcline and quartz in the natural samples, we performed freezing experiments with 2 wt % suspensions of microcline and 1 wt % suspensions of quartz reference samples in salt solutions of NaCl, CaCl₂, and MgCl₂ to mimic the saline environment in LUP. Moreover, we also varied pH and analyzed the thermograms with respect to $T_{het}$, $F_{het}$ and the melting point depression as summarized in Figs. 8 and 9.

Salt concentrations of 0.01 M, 0.1 M and 0.4 M were selected, as they comprise the salinity range of the soil and dust samples. The lowest concentration is representative for samples with EC lower than 1 dS m$^{-1}$ (Soil Sa and samples after salt removal); 0.1 M for samples with EC values of around 10 dS m$^{-1}$, and 0.4 M for samples with the highest salinity of around 40 dS m$^{-1}$ (Soil and Dust Jab). pH values were chosen to be 5 to represent the acidic conditions in soils in the absence of carbonates (samples after carbonate removal); pH 7 as typical value for slightly carbonaceous conditions, and pH 8 to represent the basic environment in calcareous soils with high content of carbonates (i.e. all samples before carbonate removal).

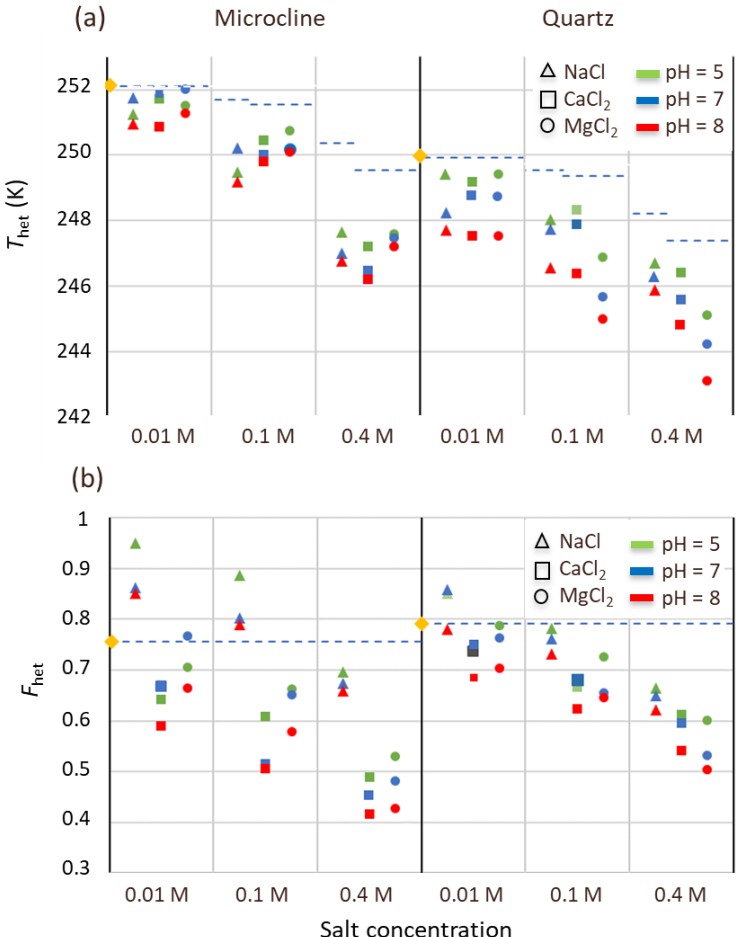

**Figure 8.** (a) Heterogeneous freezing temperatures ($T_{het}$, upper panel) and (b) $F_{het}$ (lower panel) of microcline (2 wt %) and quartz (1 wt %) in aqueous solutions of the salts NaCl (triangles), CaCl₂ (squares), and MgCl₂ (circles) at the salt concentrations 0.01 M, 0.1 M, and 0.4 M, and for pH 5 (green), pH 7 (blue), and pH 8 (red). Yellow diamonds represent $T_{het}$ and $F_{het}$ values for pure microcline (2 wt %) and quartz (1 wt %) in Milli-Q water.

415

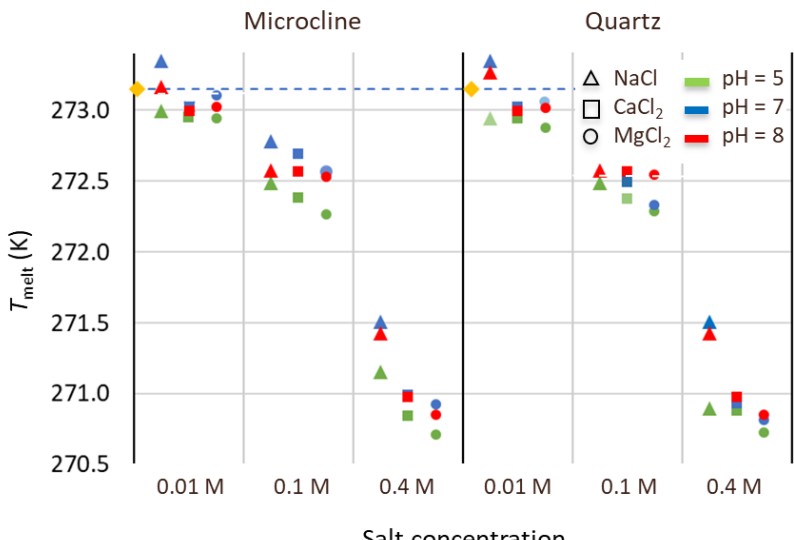

**Figure 9.** Melting points of microcline and quartz suspensions in aqueous solutions of the salts: NaCl (triangles); CaCl$_2$ (squares); MgCl$_2$ (circles) at three salt concentrations (0.01 M, 0.1 M, 0.4 M); and for three pH values: pH 5 (green); pH 7 (blue); pH 8 (red). Reference measurements with microcline and quartz suspensions (yellow diamonds) in Milli-Q water yield the melting temperature of pure water (dashed line).

Results for $F_{het}$ and $T_{het}$ of the microcline and quartz samples depending on the solution composition are presented in Fig. 8 together with the reference value for microcline and quartz in pure water. Based on these experiments, the IN activity of both quartz and microcline decreases with increasing concentration of all three alkali halides. With increasing salt concentration, a decrease in freezing onset temperatures as a function of water activity as described by the "water-activity criterion" (Koop et al., 2000; Zobrist et al., 2008) is expected, and can be derived from the melting point depression as $T_{het}(a_w) = T_{melt}(a_w + \Delta a_{w,het})$. Figure 9 shows that the melting points of the salt solutions decrease to slightly below 271 K for the highest investigated salt concentration of 0.4 M. Table 6 compares the measured melting point depression $T_{melt}$ (DSC) with the ones based on water activities calculated with AIOMFAC, $T_{melt}(a_w)$, (Zuend et al., 2008; 2010). From these data, the freezing point depression $\Delta T_{het}(a_w)$ can be derived that is expected in the absence of specific interactions between the IN active surface and the solute (Klumpp et al., 2022), which are given in Fig. 8 as blue dashed lines. Comparison of the calculated freezing point depression with the actual reduction in onset freezing temperature shown in Fig. 8 shows a larger freezing point depression than predicted based on the water-activity criterion for microcline and quartz. A decrease of freezing temperatures in microcline samples with increasing salt concentrations that exceeds the prediction based on the water-activity criterion is in agreement with findings from ice nucleation studies with K-feldspars in the presence of monovalent cations (Kumar et al., 2018; Whale et al., 2018; Yun et al., 2020). Figure 8 shows now that divalent cations have a similar effect as monovalent ions. For the investigated concentration and pH range, the cation concentration is more influential for $T_{het}$ of microcline than the nature of the cation. At pH 8, the freezing onset temperatures are only slightly reduced compared with neutral conditions. Yet, this decrease may amplify with increasing pH. For quartz, $T_{het}$ is less influenced by the solution concentration than in the case of microcline, but it is more sensitive to the nature of the solute with MgCl$_2$ inducing a stronger shift to colder temperatures than NaCl and CaCl$_2$. Moreover, for the investigated pH range, the dependence of $T_{het}$ on pH


is stronger for quartz than for microcline. The decrease in IN activity of quartz with increasing pH is in accordance with findings of Kumar et al. (2019a), and explicable by the increasing dissolution of quartz with increasing pH, which destroys nucleation sites.

Figure 8b shows that increasing the solute concentration also affects $F_{het}$ of microcline. Here, the valence of the cation seems to play a role, as the decrease in $F_{het}$ is stronger in the presence of the divalent $Ca^{2+}$ and $Mg^{2+}$ ions than for $Na^+$. In case of $Na^+$, concentrations up to 0.1 M even lead to an increase in $F_{het}$, which is in accordance with the enhanced IN activity observed by Perkins et al. (2020) in dilute NaCl solutions. Similar trends are also present for quartz but less pronounced.

**Table 6**. Water activities ($a_w$) of the salt solutions calculated with AIOMFAC (Zuend et al., 2008; 2010); melting temperatures, $T_{melt}$ ($a_w$), calculated as a function of $a_w$ with the parameterization by Koop et al. (2000) in comparison with the measured melting temperatures $T_{melt}$ (DSC) (average of quartz and microcline samples at all pH values), and corresponding freezing point depression as a function of $a_w$ derived with $T_{het}$ ($a_w$) = $T_{melt}$ ($a_w + \Delta a_{w,het}$) with $\Delta a_{w,het} = 0.2$ (Zobrist et al., 2008).

| | NaCl | | | CaCl$_2$ | | | MgCl$_2$ | | |
|---|---|---|---|---|---|---|---|---|---|
| | 0.01 M | 0.1 M | 0.4 M | 0.01 M | 0.1 M | 0.4 M | 0.01 M | 0.1 M | 0.4 M |
| $a_w$ | 0.9997 | 0.9966 | 0.9869 | 0.9995 | 0.9954 | 0.9808 | 0.9995 | 0.9954 | 0.9808 |
| $T_{melt}$ ($a_w$) (K) | 273.08 | 272.8 | 271.8 | 273.06 | 272.7 | 271.2 | 273.06 | 272.7 | 271.2 |
| $T_{melt}$ (DSC) (K) | 273.18 | 272.6 | 271.3 | 272.99 | 272.5 | 270.9 | 273.00 | 272.4 | 270.8 |
| $\Delta T_{het}$ ($a_w$) (K) | 0.04 | 0.45 | 1.76 | 0.06 | 0.61 | 2.60 | 0.06 | 0.61 | 2.60 |

Freezing onset temperatures of the samples after removal of salts, carbonates, and organic matter ($T_{het(SR+CR+OMR)}$) are on average 248.6 K for the 5 wt % samples and 247.7 K for the 2 wt % samples compared with ($T_{het(OMR)}$) of 246.7 K for both, 5 wt % and 2 wt % samples after organics removal alone. The reference freezing measurements with quartz and microcline show that, based on the pH and salt content of the LUP samples, the higher freezing temperatures of SR + CR + OMR treated samples compared with OMR treated samples is in accordance with recovered IN activity of quartz and microcline after soluble salt and carbonate removal.

Comparison with the effect of SR + OMR removal shows that the recovery in $T_{het}$ is mainly due to carbonate removal and indicates a high sensitivity of IN active sites towards pH. The on average higher $F_{het}$ of the SR + CR + OMR samples (0.72 as the average of 5 wt % and 2 wt % samples) compared with $F_{het}$ of OMR samples (0.65) is in a range that can also be ascribed to the recovered IN activity of microcline and quartz after removal of soluble salts and carbonates. These findings together with the slightly positive correlation of $T_{het}$ with quartz and microcline content of the samples after SR + CR + OMR treatments can be considered as

sufficient evidence that these minerals influence the heterogeneously frozen fraction and might even dominate freezing onset temperatures in the SR + CR + OMR treated samples. Yet, the data are less conclusive regarding the respective contributions of either quartz or microcline to the IN activity of the LUP samples. Identifying quartz as a contributor to the IN activity of the SR + CR + OMR treated samples would be among the first evidence that atmospheric quartz surfaces are sufficiently defectuous to act as a relevant atmospheric ice nucleator. The higher quartz (14.3–32 %) than microcline (3.5–6.9 %) content would speak for quartz

as the dominating INP. Specifically, the quartz concentration in the suspensions varies from 0.3–0.6 wt % in the 2 wt % samples to 0.7–1.6 wt % in the 5 wt % samples. The reference quartz sample (SA) at a suspension concentration of 1 wt % shows $F_{het}$ =





0.68 and even 0.79 for the freshly milled sample (Kumar et al., 2019a). In comparison, the microcline concentration in the suspensions varies from 0.07 to 0.14 wt % for the 2 wt % samples and from 0.18 to 0.35 wt% for the 5 wt % samples, while Kumar et al. (2018) found $F_{het}$ = 0.21 for a 0.2 wt % microcline suspension. The freezing onset temperatures of SR + CR + OMR treated

samples are about 3–4 K lower than typical values of microcline emulsions in pure water (251–252 K; Kaufmann et al, 2016; Kumar et al., 2018; Klumpp et al., 2022), but in the range observed for emulsion freezing experiments with quartz (247–250 K; Kumar et al., 2019a). Thus, the higher quartz concentration and the better agreement of the freezing onset temperatures of the LUP samples with the one of quartz than microcline, would speak for quartz as the more relevant INP in the LUP samples. Yet, the evidence is not sufficient to exclude the opposite.

**3.3 General discussion and conclusion**

The investigated soil and dust samples proved to be complex mixtures of IN promoting and inhibiting agents. Through selective removal of some components and correlations between IN activity and mineralogical composition and physicochemical properties, we have been able to identify the following main actors:

- The relatively small share of organic matter (1–5.3 %) provides the INPs that freeze at the highest temperatures in most
samples with $T_{het}$ = 248.2 K as the average of 5 wt % and 2 wt % concentrations. We inferred this finding from the reduction in $T_{het}$ after organic matter removal ($\Delta T_{het\ (OMR\text{-}natural)}$ = -1.1 K) and from the positive correlation between organic matter content and $T_{het}$ ($cc$ = 0.68). Conversely, $F_{het}$ is on average the same before and after OMR treatments, which indicates that organic INPs are substituted by mineral INPs present in the samples, which nucleate ice with a similar efficiency, yet, at lower temperatures.

The (bio-)organic INPs seem to be carbohydrates such as cellulose and/or proteinaceous molecules rather than aromatic compounds such as lignins as evidenced by the better correlation of $T_{het}$ of the natural samples with the easy-oxidizable fraction (250–350°C) than with the thermally more stable fraction (350–550°C) in TGA-MS. This finding is in accordance with studies by Hill et al. (2016) and Paramonov et al. (2018). Moreover, the organic matter seems to be associated with the clay mineral fraction, as evidenced by the strong correlation between total clay mineral content and OM content from

wet oxidation ($cc$ = 0.953). Indeed, (bio-)organic molecules are known to adsorb on clay mineral edges and basal surfaces (Kleber et al., 2021).

In most samples, the presence of soluble salts and carbonates seems to hamper the IN activity of the (bio-)organic INPs. This can be seen best for Soil Sa, in which (bio-)organic INPs dominate $T_{het}$, as evidenced by the strong reduction in the onset freezing temperature after organic matter removal. The freezing onset of this sample shifts to clearly higher

temperatures after salt and carbonate removal, showing that these components hampered the IN activity of the organic INPs, either through direct interaction with the solid mineral or via dissolved ions or a combination of both. Some samples show an increase in $F_{het}$ after organic matter removal pointing again to a direct interaction between organic matter and minerals, this time in form of an inhibiting effect of organic matter on the IN activity of the minerals. Overall, the IN activity of the organics seems to be influenced by inorganic species and vice versa. Further work is required to investigate

to what extent such interactions enhance or inhibit the IN activity of organics and minerals.

- Clay mineral content of the investigated samples ranges from 14 to 38 % and consists of kaolinite (2.2–8.4 %), smectites (0–10.6 %), polygorskite (3.8–12.6 %), and chlorite (3.9–16.8 %). This mineral class has proven to be IN active with



onset freezing temperatures of $T_{het}$ = 239–242 K for kaolinite and $T_{het}$ =239–247 K for montmorillonite in emulsion freezing experiments (Pinti et al., 2018; Kumar et al., 2019b). The other clay minerals present in the LUP samples have not yet been tested in IN experiments, but, due to their structural similarity with kaolinite and montmorillonite, most likely contribute also to IN activity. The significant correlation of total clay mineral content with $T_{het}$ ($cc$ = 0.72 as the average of 5 wt % and 2 wt % samples) and $F_{het}$ ($cc$ = 0.76) suggests a large contribution of this mineral class to the IN activity of the LUP samples, but may also just be a co-correlation paralleling the high correlation between organic matter and IN activity.

After organic matter removal, the correlation of total clay minerals with $T_{het}$ decreased only slightly ($cc$ = 0.66), although $T_{het}$ exhibited a clear downward shift from a mean value of 247.8 K to 246.7 K. Although the mean $F_{het}$ remained almost the same, its correlation with the clay mineral content decreased drastically but remained positive ($cc$ = 0.36). As the correlation of IN activity with all other minerals is even lower, clay minerals seem indeed the most dominant INP class after organic matter removal. This is supported by the heterogeneous freezing range (236–248 K) of the LUP samples after OMR removal, which agrees with the freezing temperatures of clay minerals.

-   The K-feldspar (microcline) content of the investigated samples ranges from 3.5–6.9 %. Microcline has been shown to be a highly efficient INP with freezing onset temperatures $T_{het}$ = 251–252 K in emulsion freezing experiments (Kumar et al., 2018; Klumpp et al., 2022). Yet, it seems to be irrelevant as an INP in the natural LUP samples, since their microcline content is clearly negatively correlated with both $T_{het}$ ($cc$ = -0.48 as the average of 5 wt % and 2 wt % samples) and $F_{het}$ ($cc$ = -0.51). Moreover, $T_{het}$ of most natural samples is too low for microcline suspended in pure water and further decreases after organic matter removal. The irrelevance of K-feldspar as an INP in the investigated samples is opposite to findings from previous freezing experiments with desert dust samples, which concluded that K-feldspars are relevant contributors to the observed IN activity of those samples (O'Sullivan et al., 2014; Boose et al., 2016; 2019; Paramonov et al., 2018).

Only after removal of soluble salts, carbonates, and organic matter, a positive correlation of microcline with $T_{het}$ ($cc$ = 0.31 as the average of 5 wt % and 2 wt % samples) emerged while the correlation with $F_{het}$ remained negative ($cc$ = -0.31). Moreover, the onset freezing temperatures of the thus treated samples (mean $T_{het}$ = 248.1 K) remained below the typical freezing onset temperatures of microcline.

Reference emulsion freezing experiments with a microcline suspension in salt solutions covering the concentration and pH range of the LUP samples revealed that both, the high salt concentration and pH of the natural samples hampers the IN activity of microcline. The increase in $T_{het}$ between the OMR and the SR + CR + OMR treated samples ($\Delta T_{het\ ((SR+CR+OMR)-OMR)}$ = 1.4 K as the average of 5 wt % and 2 wt % samples) suggests that mineral components within the samples can recover their IN activity after salt and carbonate removal. If this mineral component were microcline, the recovery would not be complete, as the onset freezing temperature remained below the values from emulsion freezing experiments with microcline reference samples in pure water.

In addition to the sensitivity of microcline towards soluble salts and pH, its minor concentration in the LUP samples relative to the larger contributions of other IN active species might have obscured its contribution to IN activity. In other studies, the K-feldspar concentration and its contribution to the total of IN active species showed more variation between samples. In Paramonov et al. (2018), the K-feldspar fraction varied from 1.9 to 9.3 %, and in O'Sullivan et al. (2014)


from 2.1–11 %. The microcline concentration in the samples investigated by Boose et al. (2016) ranged between 0 and 3.9 wt % with the exception of one sample with a microcline content of 30 wt %, which might have been decisive for the positive correlation between IN activity and microcline content.


- Another IN active mineral present in the LUP samples is quartz with concentrations ranging from 14 to 32 %. As the IN activity of quartz has been shown to depend on defects that can be introduced through milling (Zolles et al., 2015; Kumar et al., 2019a), it is unclear to what extent natural quartz samples are IN active. To avoid a high bias in IN activity through milling, we just sieved the soil samples to isolate the <63 μm particle fraction. In emulsion freezing experiments, freezing

onset temperatures of quartz ranged from 247 to 250 K (Kumar et al., 2019a), which coincides with the $T_{het}$ range of the untreated LUP samples. Yet, the correlation of quartz content with $T_{het}$ ($cc = -0.37$ as the average of the 5 wt % and 2 wt % samples) and $F_{het}$ ($cc = -0.39$) are both negative. This is opposite to findings by Boose et al. (2016; 2019) who found a positive correlation between quartz content and IN activity of desert dust samples, yet, some of their samples have been milled prior to performing freezing experiments.

After removal of soluble salts, carbonates, and organic matter, the correlation of quartz content with $T_{het}$ ($cc = 0.24$ and $F_{het}$ ($cc = 0.18$) turned positive for the 5 wt % samples but remained negative for the 2 wt % samples with correlations of $cc = -0.36$ for $T_{het}$ and $cc = -0.26$ for $F_{het}$. Reference emulsion freezing experiments with a quartz suspension with salt concentration and pH in the range of the LUP samples showed that, similar to microcline, also the IN activity of quartz is reduced in this environment. Thus, the recovery of quartz INPs provides a second explanation for the higher average IN

activity of SR + CR + OMR compared with OMR treated samples. The higher quartz than microcline content in the LUP samples together with the better agreement of freezing onset temperatures of reference quartz samples with $T_{het(SR+CR+OMR)}$ suggest that quartz contributes more to the IN activity of the LUP samples than microcline, yet, also the opposite assumption that microcline is the INP responsible for the increase in IN activity after SR + CR removal from the OMR sample cannot be excluded. Thus, the LUP dataset is not able to answer the question whether natural quartz is a relevant

contributor to IN activity in the atmosphere.

- Soluble salts may be present as crystalline phases and/or amorphous brine. As they are soluble, we can exclude them as INPs, but they may interfere with the IN activity of other INPs. Seven out of the eight LUP samples contain halite (crystallized NaCl) at concentrations from 1.3 to 5.7 %, which completely dissolves in 5 wt % and 2 wt % aqueous

suspensions. The low correlation between halite content and electrical conductivity ($cc = 0.21$) indicates the presence of amorphous salts in addition, likely chlorides, sulfates, carbonates and bicarbonates of $K^+$, $Ca^{2+}$, and $Mg^{2+}$ as these are the most abundant ions in LUP sediments (Hamzehpour et al., 2022; Sharifi et al., 2018). Dissolved salts are known to decrease freezing temperatures as a function of water activity according to the water-activity criterion (Koop et al., 2000; Zobrist et al., 2008). Yet, reference freezing experiments with quartz and microcline evidenced a decrease in $T_{het}$ and $F_{het}$

that exceeds the expected effect of water activity for both minerals. The negative correlation of electrical conductivity with $T_{het}$ ($cc = -0.63$ as the average of 5 wt % and 2 wt % samples) and $F_{het}$ ($cc = -0.60$) further supports the hampering effect the high salinity has on the IN activity of the LUP samples. As the impact of soluble salts on IN activity depends on the suspension concentration, a higher dilution should reduce it. This can be seen for $F_{het}$ of the untreated Dust Jab, which is the sample with the highest electrical conductivity (43.9 dS.m$^{-1}$): while $F_{het}$ of the 2 wt % sample is 0.62, the one



of the 5 wt % sample is only 0.49. At even lower concentration, the effect of soluble salts should decrease even more and might become negligible for immersion freezing in cloud droplets.

- Based on the XRD analysis, total carbonate concentrations vary between 13.1 and 28.5 % in the LUP samples with contributions of calcite (9.7–13.5 %), dolomite (1.9–4.5 %), aragonite (0–13.4 %), and magnesite (0–4.9 %). As calcite

and dolomite did not show any IN activity in emulsion freezing experiments (Kaufmann et al., 2016), we consider the IN activity of all carbonate minerals as negligible. The irrelevance of carbonates as INPs is further supported by the negative correlation of total carbonates with $T_{het}$ ($cc$ = -0.23 as the average of 5 wt % and 2 wt % samples) and $F_{het}$ ($cc$ = -0.40). Rather, this negative correlation together with the increase in $T_{het}$ ($\Delta T_{het\ (CR-natural)}$ = 1.2 K) and $F_{het}$ ($RT_{het\ (CR/natural)}$ = 1.2) after carbonate removal suggest that carbonates inhibit the INPs present in the samples. This hampering effect could occur

by occlusion of active sites through direct interaction with the minerals or through release of the ions $Ca^{2+}$, $Mg^{2+}$, and $CO_3^{2-}$ to the solution. Release of $CO_3^{2-}$ is responsible for the basic pH of the suspensions, which may influence the IN activity of INPs through deprotonation of IN active sites or by increasing the solubility of minerals, as is e.g. the case for quartz (Kumar et al., 2019b). Reference measurements showed indeed that the IN activity of quartz and microcline is reduced already at pH 8 compared to neutral conditions. Yet, the anticorrelation of IN activity with pH is less pronounced

than the one with total carbonates, that is, the correlation of pH with $F_{het}$ of the natural samples is only slightly negative ($cc$ = -0.25 as the average of 5 wt % and 2 wt % samples), but slightly positive with $T_{het}$ ($cc$ = 0.19). As the freezing onset of the natural samples is dominated by INPs originating from organic matter, the positive correlation of $T_{het}$ with pH compared to the negative one with carbonates makes it unlikely that the inhibition of (bio-)organic INPs originates from a pH effect. Rather, it might be due to direct interactions with the minerals either through occlusion of (bio-)organic INPs

from the solution through a cementing effect, by their adsorption on the mineral surface, or through interactions with dissolved $Ca^{2+}$ which can form inner sphere and/or outer sphere complexes with organic matter (Rowley et al., 2018). Further studies are required to elucidate; which interactions are most relevant for the reduced IN activity.

In summary, the analysis of the effects that the different treatments exert on the IN activity of the LUP samples revealed complex interactions between organic matter and mineral particles in soils. This study was able to elucidate some of these interactions, yet,

more research is required to disentangle how the mixing state of organic and inorganic soil components influence their ability to nucleate ice. While research over the past years established feldspars, quartz, and clay minerals as the most relevant mineral INPs in dust, less is known about the composition of the (bio-)organic matter that enhances the IN activity of fertile soils above the one of mineral dust. More research is required to elucidate the interactions between organic matter and mineral particles to improve our understanding of the origin of high IN activity in fertile soils.

*Data availability.* The data presented in this publication will be submitted to the ETHZ data repository soon.

*Author contributions.* NH conducted the experiments and field works. KK provided equipment and facilities for lab experiments. NH, CM, and TP contributed to the planning and interpretation of the experiments. NH and CM prepared the manuscript with contributions from TP and KK. DT conducted the TGA analysis.

*Competing interests.* The authors declare that they have no conflict of interest.

*Acknowledgements.* We thank Michael Plötze for XRD measurements; Ulrich Krieger and Uwe Weers for support in the laboratory.



*Financial support.* NH has been supported by visiting professors grant at ETH. KK has been supported by the Swiss National Foundation (project number 200021_175716).

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



**Appendix**

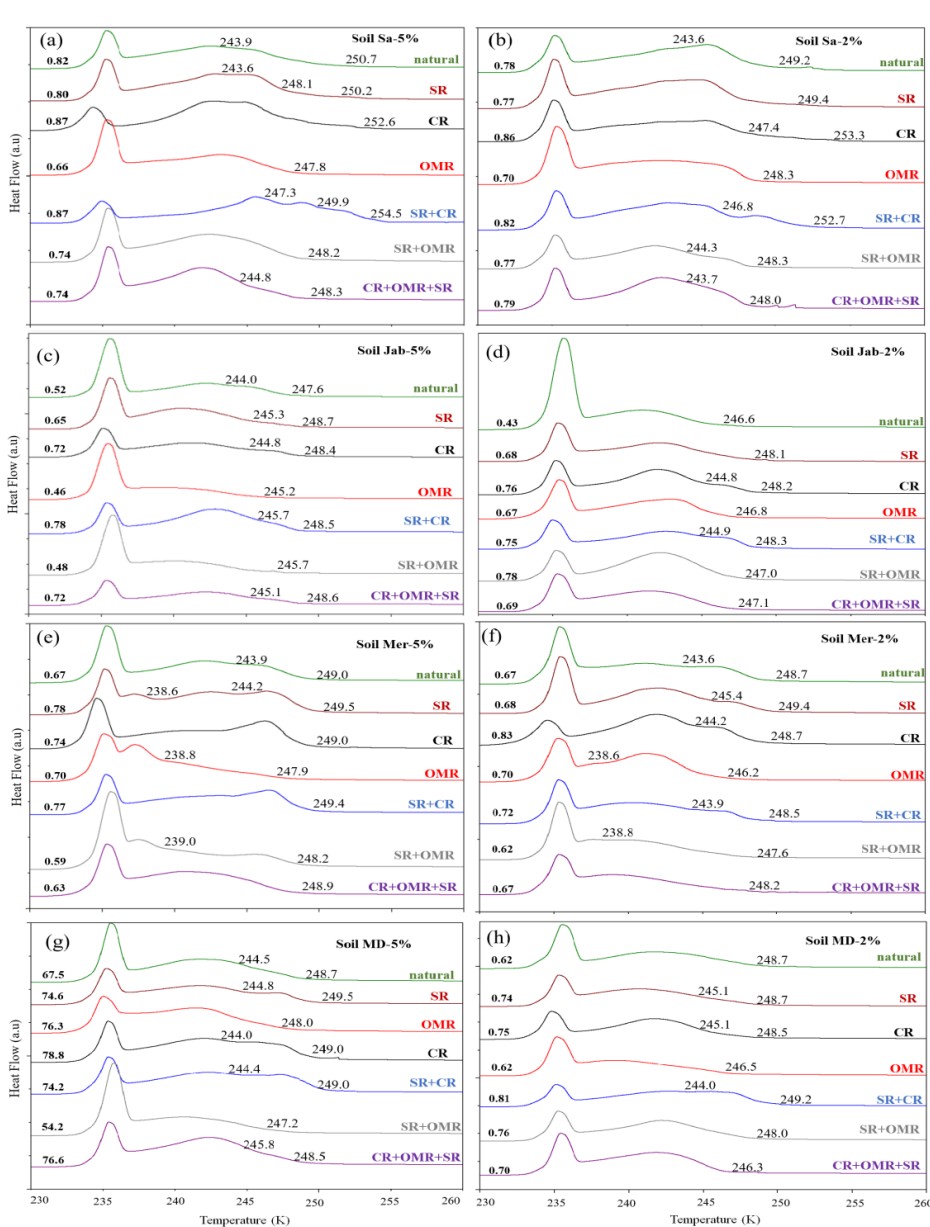

**Figure A1:** DSC thermograms of treated soil samples with 5 wt % (left panels) and 2 wt % (right panels) concentrations in comparison with the untreated samples. SR: salt removed; CR: carbonate removed; OMR: organic matter removed; SR + CR: salt and carbonate removed; SR + OMR: salt and organic matter removed; SR + CR + OMR: salt, carbonate and organics removed. $T_{het}$ in Kelvin and $F_{het}$ in fraction are displayed directly on the curves.

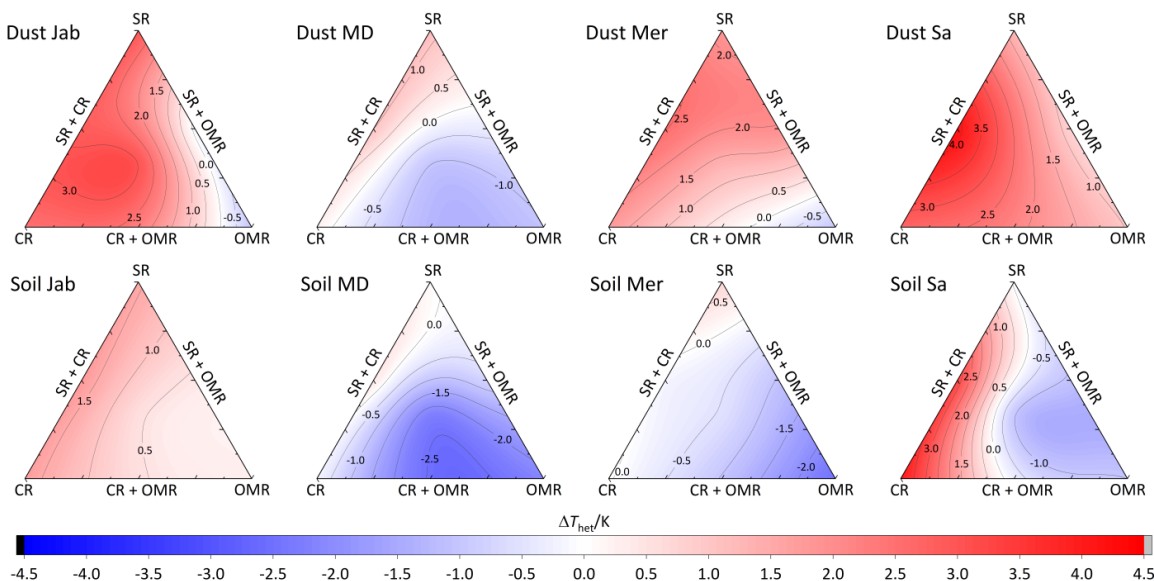

**Figure A2:** Effects of treatments in terms of $\Delta T_{het\,(treatment-natural)}$ for dust and soil samples at 2 wt % concentration. SR: salt removed; CR: carbonate removed; OMR: organic matter removed; SR + CR: salt and carbonate removed; SR + OMR: salt and organic matter removed; SR + CR + OMR: salt, carbonate and organic removed treatments.


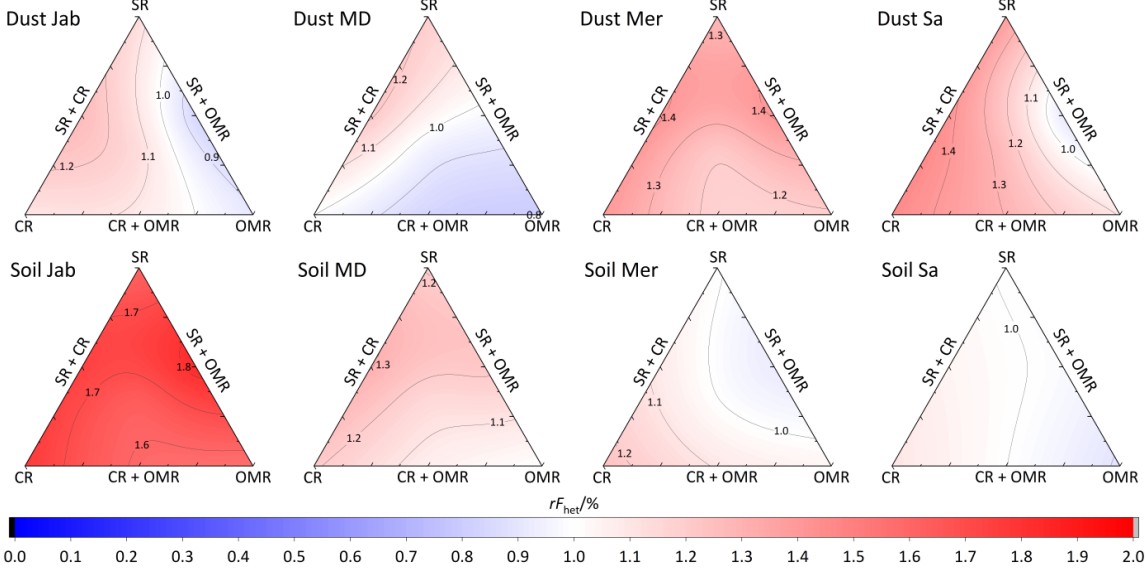

**Figure A3:** Effects of treatments in terms of $rF_{het\,(treatment-natural)}$ for dust and soil samples at 2 wt % concentration. SR: salt removed; CR: carbonate removed; OMR: organic matter removed; SR + CR: salt and carbonate removed; SR + OMR: salt and organic matter removed; SR + CR + OMR: salt, carbonate and organic removed treatments.





**Figure A4:** TG (green) and DSC (blue) curves of soil and dust samples with corresponding mass data plotted against temperature. Light blue: total ion current of mass 44 ($CO_2$)