# Peer review of "The Urmia Playa as a source of airborne dust and ice nucleating particles – Part 2: Unraveling the relationship between soil dust composition and ice-nucleation activity"

_Atmospheric Chemistry and Physics, 2022_

## Referee Comment (RC1)

Review Hamzehpour et al. (2022): The Urmia Playa as source of airborne dust and ice nucleating particles – Part 2: Unraveling the relationship between soil dust composition and ice-nucleation activity

**Summary**

The manuscript submitted by Nikou Hamzehpour and co-authors provides a detailed insight into the potential driving factors of the ice-nucleation (IN) ability of dust particles from a dried lakebed. Due to an increase in desertification, it is expected that such sources for airborne dust particles are becoming more abundant, and might therefore impact cloud microphysical processes such as the initiation of ice crystal formation.

Soil dust particles can have distinct IN abilities related to their mineralogical composition, and they can contain organic matter and soluble salts. Here, the authors investigate the impact of each component on the ice nucleation ability of the dust sample. The bio-organic matter is found to determine the onset temperature for ice nucleation, and the removal of soluble salts and carbonates leads to an increase in the IN activity. After the removal of the constituents, the IN activity is determined by the clay mineral fraction, and to a lesser extent to quartz and microcline.

The paper is well written and I only have minor comments and suggestions. More of such systematic investigations of driving factors for the ice nucleation ability of natural soil samples are needed to improve our understanding in this field.

**General comments**

- Please consider shortening the abstract.
- I recommend moving some figures in the appendix and only showing key figures in the manuscript (e.g., figures 3, 4, 10).
- In some cases, the mentioned publications are examples and do not represent all existing literature. Please check and make use of "e.g." in such cases or complete the cited literature.
- What is the atmospheric relevance regarding the size of the samples collected with the high-volume samplers and the ground-collected samples? Supermicron particles are typically not transported over longer distances, such that they could be lofted into levels in the atmosphere where they could impact cloud microphysics by their ice nucleation ability. Please elaborate on this.
- Is it possible to give a temperature-dependent $F_{het}$? This might allow comparing the nucleation efficiency of the organic INPs and the dust INPs in the sample.
- What is the temperature uncertainty of the DSC experiments? E.g., in line 284, you state a value of -0.2 K for $\Delta T_{het}$ which could be within the uncertainty of the experiment. What are significant changes in values for $\Delta T_{het}$ and $rF_{het}$?
- Are there studies investigating the impact of carbonate removal and salt removal on the ice nucleation ability of dust particles, or is your study the first one investigating this?
- Lines 336 – 337: It might be interesting to the reader to compare your results in a more quantitative way to these studies.
- Line 533: The heterogeneous freezing temperature range of 236 – 248 K is very broad and includes not only the freezing temperatures of clay minerals but also other mineral types.

**Technical comments**

- Title: Should it not be „… as **a** source…"?
- Line 163: It should be mentioned in the table header that it is taken from the first part of this work (Hamzehpour et al., 2022).
- Line 251: Abbreviation „$rF_{het}$" is not explained.

- Line 310: I recommend increasing the marker size and to indicate the dust and soil samples with different markers.
- Figure 11: I assume that the blue dashed line corresponds to $T_{het}$ and $F_{het}$ of the pure minerals, as described in Figure 12?
- Figure A is not specifically mentioned in the text. Also, the labels are too small.

---

## Author Response (AR1)

**We thank Reviewer 1 for his/her thoughtful comments. We reproduce the reviewer's comments in black and our responses in blue. Line numbers refer to the revised manuscript.**

Review of Hamzehpour et al.

This manuscript reports the outcomes of a range of experiments aimed at determining which components of a range of soil samples are responsible for their ice nucleation activity. The manuscript concludes, reasonably I think, that bio-organic matter is probably responsible for the ice nucleation activity of unaltered samples. It is found that removing carbonates and salts from the dust increases the ice nucleation activity of both the mineral and organic fractions of the samples. It is an interesting and thorough study and the paper is well written. I have a few comments which the authors may wish to consider, however I support publication.

Comments

The conclusions are based on quite small temperature shifts. I think it is important that there is some discussion of the temperature uncertainty of the two measurements reported from the DSC, as mentioned by Referee 2.

Reply: DSC has in fact a high precision. To clarify this, we add the following sentence to the manuscript on line 239:

"The average precision in $T_{het}$ is ±0.2 K. Uncertainties in $F_{het}$ are on average ±0.05, but may be much larger when heterogeneous freezing signals are weak or overlap (forming a shoulder) with the homogeneous freezing signal."

Page 19 line 438- Whale (2022) recently showed that $MgCl_2$ has a similar effect on ice nucleation by feldspar to monovalent cations.

Reply: Thanks for pointing this paper out. We added this reference to the text on line 445:

"These findings are supported by Whale et al. (2022) who recently showed that $MgCl_2$ has a similar effect on ice nucleation by feldspar as monovalent cations."

It seems relevant that the ice nucleation activities of both alkali feldspars (Harrison et al., 2016) and quartzes (Harrison et al., 2019) are known to vary a great deal. Something of why this is known for feldspars (Whale et al., 2017; Kiselev et al., 2021) however, as far as I am aware, there is no good explanation at all for the quite broad range of ice nucleation activities observed for quartz samples. It may turn out to be incorrect to treat all 'quartz' and all 'feldspar' as having the same responses to solution environment and the treatments laid out here. The 'reference' experiments are certainly interesting and worthwhile but there seems to me no guarantee at this point that the quartz and feldspar in the natural samples will behave similarly with regard to salt and pH environment.

Reply: The sensitivities of quartz and feldspars to ions and pH are indeed not the same, as we have discussed in Kumar et al. (2019a; 2019b). The decrease in IN activity of quartz at elevated pH and in the presence of ions can be traced back to the increased dissolution rate of quartz under basic

conditions and with increasing ionic strength. During quartz dissolution, the best nucleation sites, which we assume to be high-energy sites that can be introduced through milling, are lost. Because we were aware of these sensitivities from our previous work, we wanted to quantify how large the effect is under the pH and ionic strength conditions covered by the LUP samples.

We discuss the sensitivities of feldspars and quartz to pH and ions in the manuscript from lines 440–451. We also refer to Kumar et al. (2019a), where the IN activity of quartz is discussed in detail.

For instance the work of Perkins et al. (2020) and Yun et al. (2020) showed unexplained enhancements of ice nucleation by feldspar in the presence of $K^+$ ions, which hasn't been observed in other studies, suggesting that there may be more than one mechanism responsible for ice nucleation by feldspars. In a similar vein, I would not be entirely shocked if it turned out hydrogen peroxide treatment damaged the ice nucleating ability of minerals. A recent study by Daily et al (2022) showed that heat treatment can slightly impair the ice nucleating ability of feldspar for instance, which probably wouldn't have been expected.

Reply: We are aware of Daily et al. (2022). This is indeed an important study helping to assign changes in IN activity after heat treatments to the components responsible for them. A similar study for $H_2O_2$ digestion would be welcome. Yet, our assignment of IN activity to organic matter is not only based on $H_2O_2$ digestion, but also on the significant correlation that we find between organic matter content and IN activity of the samples.

In our work, we dried the samples at $65^oC$ (dry heating). In Daily et al. (2022) temperatures above $250^oC$ were required to damage IN activity. In wet heating (e.g. while removing carbonates or organics), we kept temperature below $80^oC$, while in Daily et al. (2022) samples needed to be heated above $90^oC$ to damage IN activity.

We will add the following text to the revised manuscript on line 213:

"Based on the study by Daily et al. (2022), samples needed to be heated above $90°C$ to damage IN activity of the minerals. Therefore, we do not expect a negative effect on the IN activity of the mineral components due to the heating required to remove organic matter, yet, we cannot exclude it."

I don't think these concerns undermine the study, but I do think it would be a good idea to make clear that the complexity of the problem is such that there are unknowns, and that the various chemical treatments employed could have unintended impacts on observed ice nucleation.

References

Daily, M. I., Tarn, M. D., Whale, T. F., and Murray, B. J.: An evaluation of the heat test for the ice-nucleating ability of minerals and biological material, Atmos. Meas. Tech., 15, 2635-2665, 10.5194/amt-15-2635-2022, 2022.

Harrison, A. D., Whale, T. F., Carpenter, M. A., Holden, M. A., Neve, L., O'Sullivan, D., Vergara Temprado, J., and Murray, B. J.: Not all feldspars are equal: a survey of ice nucleating properties across the feldspar group of minerals, Atmos. Chem. Phys., 16, 10927-10940, 10.5194/acp-16-10927-2016, 2016.

Harrison, A. D., Lever, K., Sanchez-Marroquin, A., Holden, M. A., Whale, T. F., Tarn, M. D., McQuaid, J. B., and Murray, B. J.: The ice-nucleating ability of quartz immersed in water and its atmospheric importance compared to K-feldspar, Atmos. Chem. Phys., 19, 11343-11361, 10.5194/acp-19-11343-2019, 2019.

Kiselev, A., Keinert, A., Gaedecke, T., Leisner, T., Sutter, C., Petrishcheva, E., and Abart, R.: Effect of chemically induced fracturing on the ice nucleation activity of alkali feldspar, Atmos. Chem. Phys. Discuss., 2021, 1-17, 10.5194/acp-2021-18, 2021.

Perkins, R. J., Gillette, S. M., Hill, T. C. J., and DeMott, P. J.: The Labile Nature of Ice Nucleation by Arizona Test Dust, ACS Earth and Space Chemistry, 4, 133-141, 10.1021/acsearthspacechem.9b00304, 2020.

Whale, T. F., Holden, M. A., Kulak, A. N., Kim, Y.-Y., Meldrum, F. C., Christenson, H. K., and Murray, B. J.: The role of phase separation and related topography in the exceptional ice-nucleating ability of alkali feldspars, Phys. Chem. Chem. Phys., 10.1039/C7CP04898J, 2017.

Whale, T. F.: Disordering effect of the ammonium cation accounts for anomalous enhancement of heterogeneous ice nucleation, The Journal of Chemical Physics, 156, 144503, 10.1063/5.0084635, 2022.

Yun, J., Link, N., Kumar, A., Shchukarev, A., Davidson, J., Lam, A., Walters, C., Xi, Y., Boily, J.-F., and Bertram, A. K.: Surface Composition Dependence on the Ice Nucleating Ability of Potassium-Rich Feldspar, ACS Earth and Space Chemistry, 4, 873-881, 10.1021/acsearthspacechem.0c00077, 2020.

Kumar, A., Marcolli, C., and Peter, T.: Ice nucleation activity of silicates and aluminosilicates in pure water and aqueous solutions – Part 2: Quartz and amorphous silica, Atmos. Chem. Phys., 19, 6035–6058, doi:10.5194/acp-19-6035-2019, 2019a.

Kumar, A., Marcolli, C., and Peter, T.: Ice nucleation activity of silicates and aluminosilicates in pure water and aqueous solutions – Part 3: Aluminosilicates, Atmos. Chem. Phys., 19, 6059–6084, doi:10.5194/acp-19-6059-2019, 2019b.

**We thank Reviewer 2 for his/her thoughtful comments. We reproduce the reviewer's comments in black and our responses in blue. Line numbers refer to the revised manuscript.**

**Summary**

The manuscript submitted by Nikou Hamzehpour and co-authors provides a detailed insight into the potential driving factors of the ice-nucleation (IN) ability of dust particles from a dried lakebed. Due to an increase in desertification, it is expected that such sources for airborne dust particles are becoming more abundant, and might therefore impact cloud microphysical processes such as the initiation of ice crystal formation.

Soil dust particles can have distinct IN abilities related to their mineralogical composition, and they can contain organic matter and soluble salts. Here, the authors investigate the impact of each component on the ice nucleation ability of the dust sample. The bio-organic matter is found to determine the onset temperature for ice nucleation, and the removal of soluble salts and carbonates leads to an increase in the IN activity. After the removal of the constituents, the IN activity is determined by the clay mineral fraction, and to a lesser extent to quartz and microcline.

The paper is well written and I only have minor comments and suggestions. More of such systematic investigations of driving factors for the ice nucleation ability of natural soil samples are needed to improve our understanding in this field.

**General comments**

- Please consider shortening the abstract
  Reply: we shortened the abstract

- I recommend moving some figures in the appendix and only showing key figures in the manuscript (e.g. figures 3, 4, 10).

  Reply: Figures are moved to the appendix and figure numbering in the text is revised throughout the manuscript.

- In some cases, the mentioned publications are examples and do not represent all existing literature. Please check and make use of " e.g." in such cases or complete the cited literature.

  Reply: We added "e.g." where applicable.

- What is the atmospheric relevance regarding the size of the samples collected with high-volume samplers and the ground-collected samples? Super micron particles are typically not transported over longer distances, such that they could be lofted into levels in the atmosphere where they could impact microphysics by their ice nucleation ability. Please elaborate on this.

  Reply: Recent studies have shown that dust with diameters >5 μm and even >20 μm is longer lived in the atmosphere and present in larger amounts than previously assumed (e.g. Adebiyi and Kok, 2020; Heisel et al., 2021; Froyd et al., 2022).

  Moreover, droplets within emulsions contain only few particles. Since the number size distribution of the dust and soil samples peaks in the submicron region, the freezing signal in emulsion freezing experiments is characteristic for this particle class.

- Is it possible to give temperature-dependent F het? This might allow comparing nucleation efficiency of the organic INPs and dust INPs in the sample.

  Reply: The differential scanning calorimeter registers heat transfer and not freezing events. This hampers an unambiguous division between organic and mineral signal. We think that $T_{het}$ and $F_{het}$ are the most robust parameters to characterize the DSC thermograms. To give a visual impression of the freezing signal, we also depict the whole thermograms in the manuscript.

- What is the temperature uncertainty of the DSC experiments? E.g., in line 284, you state a value of -0.2 k for $\Delta T_{het}$ which could be within the uncertainity of the experiment. What are significant changes in values for $\Delta T_{het}$ or $rF_{het}$?

  Reply: DSC has in fact a high precision. To clarify this, we add the following sentence to the manuscript on line 239:

  "The average precision in $T_{het}$ is ±0.2 K. Uncertainties in $F_{het}$ are on average ±0.05, but may be much larger when heterogeneous freezing signals are weak or overlap (forming a shoulder) with the homogeneous freezing signal."

- Are there studies investigating the impact of carbonate removal and salt removal on the ice nucleation ability of dust particles or is your study the first one investigating this?

  We are not aware of other studies that investigated the effect of carbonate and salt removal. It would be interesting to see more such studies. Yet, soluble salt removal is only meaningful for salty soils, which might be rare for desert dusts.

- Lines 336-337: it might be interesting to the reader to compare your results in a more quantitative way to these studies.

  The strength of the DSC method is to give a qualitative overview over the IN activity of a sample, which is provided by the whole DSC thermograms. We decided to break their rich information down to $T_{het}$ and $F_{het}$. Another way of quantifying the curves would also have been possible. We would like to keep to this method, as we have evaluated the DSC thermograms in this way also in previous studies (e.g. Kumar et al., 2018; 2019a; 2019b).

- Line 553: the heterogeneous freezing temperature range of 236-248 k is very broad and includes not only the freezing temperature of clay minerals but also other mineral types.

  We agree with the reviewer. We therefore revise this statement:

  "This is supported by the heterogeneous freezing range (236–248 K) of the LUP samples after OMR removal, which is in general agreement with the freezing temperatures of clay minerals, although it is not specific for them as other mineral types are IN active within the same temperature range."

Technical comments

- Title: should it not be,,….. as a source?

Reply: corrected

- Line 163: it should be mentioned in the table header that it is taken from the first part of this work (Hamzehpour et al., 2022).

Reply: corrected

- Line 251: abbreviation „ rFhet" is not explained.

Reply: it is now explained in the text

- Line 310: I recommend increasing the marker size and to indicate the dust and soil samples with different markers.

- Figure 11: I assume that the blue dashed line corresponds to $T_{het}$ and $F_{het}$ of the pure minerals, as described in Figure 12?

Reply: to explain the blue dashed line, we add the following text to the figure caption of Fig. 8 (former Fig. 11):

"The blue dashed line in panel (a) represents the freezing temperature according to the water activity criterion, in panel (b), it marks the frozen fractions of the minerals in pure water."

- Figure A is not specifically mentioned in the text. Also, the labels are too small.

Reply: Figure A is now Figure A4 and is mentioned in line 354.

**References**

Adebiyi, A. A. and Kok, J. F.: Climate models miss most of the coarse dust in the atmosphere, Sci. Adv., 6, 15, doi:10.1126/sciadv.aaz9507, 2020.

Froyd, K. D., Yu, P., Schill, G. P., Brock, C. A., Kupc, A., Williamson, C. J., Jensen, E. J., Ray, E., Rosenlof, K. H., Bian, H., Darmenov, A. S., Colarco, P. R., Diskin, G. S., Bui, T., and Murphy, D. M.: Dominant role of mineral dust in cirrus cloud formation revealed by global-scale measurements, Nat. Geosci., 15, 177–183, doi:10.1038/s41561-022-00901-w, 2022.

Heisel, M., Chen, B., Kok, J. F., and Chamecki, M.: Gentle topography increases vertical transport of coarse dust by orders of magnitude, J. Geophys. Res., 126, e2021JD034564, doi:10.1029/2021JD034564, 2021.

Kumar, A., Marcolli, C., Luo, B., and Peter, T.: Ice nucleation activity of silicates and aluminosilicates in pure water and aqueous solutions – Part 1: The K-feldspar microcline, Atmos. Chem. Phys., 18, 7057–7079, doi:10.5194/acp-18-7057- 2018, 2018.

Kumar, A., Marcolli, C., and Peter, T.: Ice nucleation activity of silicates and aluminosilicates in pure water and aqueous solutions – Part 2: Quartz and amorphous silica, Atmos. Chem. Phys., 19, 6035–6058, doi:10.5194/acp-19-6035-2019, 2019a.

Kumar, A., Marcolli, C., and Peter, T.: Ice nucleation activity of silicates and aluminosilicates in pure water and aqueous solutions – Part 3: Aluminosilicates, Atmos. Chem. Phys., 19, 6059–6084, doi:10.5194/acp-19-6059-2019, 2019b.